



# SMAUG v1.0 – a user-friendly muon simulator for transmission tomography of geological objects in 3D

Alessandro Lechmann[1], David Mair[1], Akitaka Ariga[2], Tomoko Ariga[3], Antonio Ereditato[2], Ryuichi Nishiyama[4], Ciro Pistillo[2], Paola Scampoli[2,5], Mykhailo Vladymyrov[2] and Fritz Schlunegger[1]

[1]Institute of Geological Sciences, University of Bern, Bern, CH-3012, Switzerland
[2]Albert Einstein Center for Fundamental Physics, Laboratory for High Energy Physics, University of Bern, Bern, CH-3012, Switzerland
[3]Faculty of Arts and Science, Kyushu University, Fukuoka, JP-819-0385, Japan
[4]Earthquake Research Institute, The University of Tokyo, Tokyo, JP-113-0032, Japan
[5]Physics Department "Ettore Pancini", University of Naples Federico II, Naples, IT-80126, Italy

*Correspondence to*: Alessandro Lechmann (alessandro.lechmann@geo.unibe.ch)

**Abstract.**

Knowledge about muon tomography has spread in recent years in the geoscientific community and several collaborations between geologists and physicists have been founded. As the data analysis is still mostly done by particle physicists, we address the need of the geoscientific community to participate in the data analysis, while not having to worry too much about the particle physics equations in the background. The result hereof is SMAUG, a toolbox consisting of several modules that cover the various aspects of data analysis in a muon tomographic experiment. In this study we show how a comprehensive geophysical model can be built from basic physics equations. The emerging uncertainties are dealt with by a probabilistic formulation of the inverse problem, which is finally solved by a Monte Carlo Markov Chain algorithm. Finally, we benchmark the SMAUG results against those of a recent study, which however, have been established with an approach that is not easily accessible to the geoscientific community. We show that they reach identical results with the same level of accuracy and precision.

## 1 Introduction

Among the manifold geophysical imaging techniques, muon tomography has increasingly gained the interest of geoscientists during the course of the past years. Before its application in Earth sciences, it was initially used for archaeological purposes. Alvarez et al. (1970) used this method to search for hidden chambers in the pyramids of Giza, in Egypt; an experiment which was recently repeated by Morishima et al. (2017), as better technologies have continuously been developed. Other civil engineering applications include the monitoring of nuclear power plant operations (Takamatsu et al., 2015) and the search for nuclear waste repositories (Jonkmans et al., 2013) as well as the investigation of underground tunnels (e.g. Thompson et al., 2020; Guardincerri et al., 2017). A serious deployment of muon tomography in Earth sciences has only begun in the past





decades. These undertakings mainly encompass the study of the interior of volcanoes in France (Ambrosino et al., 2015; Jourde et al., 2016; Noli et al., 2017; Rosas-Carbajal et al., 2017), Italy (Ambrosino et al., 2014; Lo Presti et al., 2018; Tioukov et al., 2017), and Japan (Kusagaya and Tanaka, 2015; Nishiyama et al., 2014; Oláh et al., 2018; Tanaka, 2016).

Other experiments have been performed in order to explore the geometry of karst cavities in Hungary (Barnaföldi et al., 2012) and Italy (Saracino et al., 2017). Further studies (see also review article by Lechmann et al., 2021a) have been conducted by our group to recover the ice-bedrock interface of Alpine glaciers in central Switzerland (Nishiyama et al., 2017; 2019).

The core component of every geophysical exploration experiment is formed by the inversion, which might be better known

to other communities as fitting or modelling. This is where the model parameters are found, which best fit the observed data. Up until now, this central part has mostly been built specifically to meet the needs of the experimental campaign at hand. On the one hand this approach has the advantage of allowing the consideration of the peculiarities of particle detectors, their data processing chain, and other models involved (e.g. the cosmic ray flux model). On the other hand, when every group develops a separate inversion algorithm, the reconstruction of the precise calculations performed in the data analysis procedure

becomes a challenge. For a researcher who is not familiar with the intricacies of inversion, this might even be tougher. We thus see the need for a lightweight programme that incorporates a structured and modular approach to inversion, that also allows users with little inversion experience to familiarise themselves with this rather involved topic. This programme can be used to directly analyse experimental data in a stand-alone working environment, and the modules and theoretical foundations can be adapted, customised, and integrated into new programmes. For this reason, the code is built in the

programming language Python as to facilitate the exchange between researchers and to enhance modifiability. Moreover, the source code is freely available online (see code availability section below).

To facilitate the further reading of our code, we introduce the reader at this point to our benchmark experiment, to which we will refer on multiple occasions throughout this work. The experimental campaign is explained in detail in Nishiyama et al. (2017) and thus we will resort to a description of the experimental design at this point. In the Nishiyama et al. (2017) study

we aimed at we aimed at recovering the ice-bedrock interface of an Alpine glacier in central Switzerland. Figure 1 shows that we had access to the railway tunnel of the Jungfraubahnen, where we installed three detectors. In our measurement we recorded muons from directions that consisted purely of rock and others where we knew that the muons must have crossed rock and ice (see the two cones in Fig. 1). From the former it was possible, together with laboratory measurements, to determine the physical parameters of the rock more precisely. Conversely, we utilised the measurement of the latter to infer

the interface between rock and ice underneath the glacier. Finally, we will also use the results of that experiment (Nishiyama et al., 2017) to verify our new algorithm in the present study.



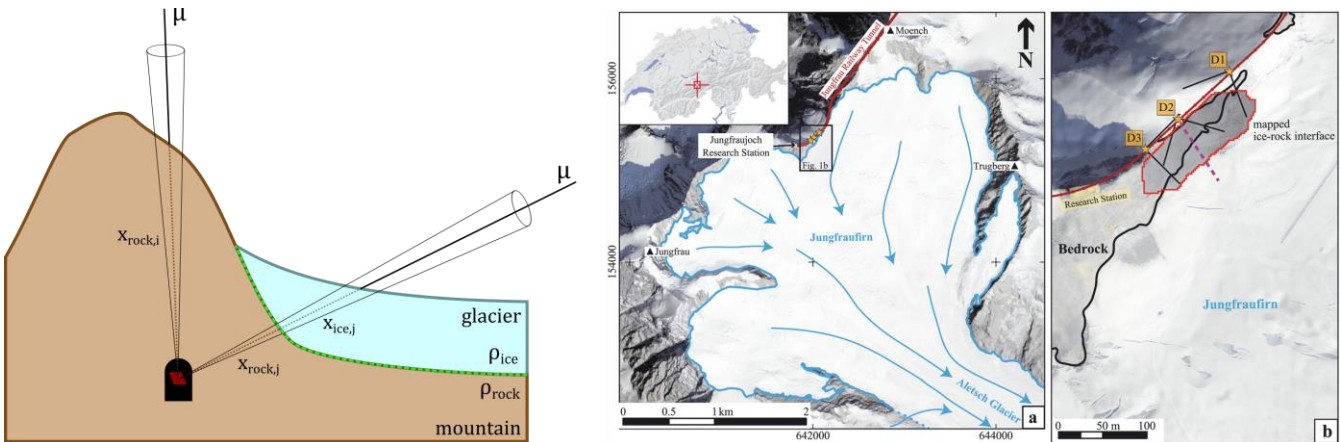

**Figure 1:** *Left:* **Schematic side view of the experiment from Nishiyama et al. (2017). A muon detector (red) is placed in a tunnel or cavity and records muons from the cosmic ray flux that penetrate the material (ice & rock) from different directions. Muons that are detected along cones yield information on the amount and density of matter between the topographic surface and the detector. Based on this the interface between ice and rock (dashed green line) can be reconstructed.** *Right:* **Fig. 1 from Nishiyama et al. (2017); Overview of the study region in the central Swiss Alps. a) shows an outline of the imaged glacier and its localisation in Switzerland and b) indicates where the three detectors have been positioned in the Jungfraubahnen railway tunnel.**

## 1.1 Inversion – a modular view

The goal of every muon tomography study is essentially to extract information on the physical parameters of the radiographed object through a measurement of the cosmic ray muon flux and an assessment of its absorption as the muons cross that object. In geological applications these objects are almost always lithological underground structures such as magma chambers, cavities, or other interfaces with a high-density contrast. The reconstruction of the geometry of such structures can only be achieved if the measured muon data is compared to the results of a muon flux simulation. As stated earlier, this is the basic principle of the inversion procedure. However, the aforementioned "muon flux simulation" is not just a simple programme, but it consists of several physically independent models that act together. Taking a modular view, we will call these models "modules" from here on, as they will inevitably be part of a larger inversion code. We have visualised the components that are necessary to build an inversion in Fig. 2.

The first of the modules is the input module for the experiment results, which also considers the detectors that were used in the experiment. Typical detector setups include nuclear emulsion films (e.g. Ariga et al., (2018), cathode chambers (e.g. Oláh et al., 2013), scintillators (e.g. Anghel et al., 2015) or other hardware solutions. Although the detailed data processing chain may be comprehensive, the related output almost always comes in the form of a measured directional (i.e. from various incident angles) muon flux, which will be the input to the inversion scheme. Here, we primarily work with the premise that the muon flux data and the associated errors are given. The corresponding errors can then be furnished to the code by means of an interface. The simulation module on the other hand, consists of four autonomous components (see Fig. 2). First, a cosmic ray muon flux model is necessary, which describes the muon abundance in the atmosphere and which is generally dependent on the muon energy, its incidence angle, and the altitude of the detector location. Lesparre et al. (2010) list and





compare various muon flux models that may be incorporated into an extensive simulation. Second, it is necessary to model

the spatial distribution of the detectors as well as the initial distribution of the lithologies. Related pre-existing software solutions mainly comprise GIS- and geological 3D-modelling applications, that excel at capturing and compiling geological information from various sources (e.g. field, maps, etc.) into a spatially organised database. Third, the lithologies consisting of different minerals have to be translated to a set of parameters, which are a necessary input for the subsequent physical simulation. This can be done by a rock model (e.g. Lechmann et al., 2018), which considers the effects of the mass density as

well as the average atomic mass and charge of the rock as a function of its mineralogical composition. Lastly, the muon fluxes at the detector sites have to be simulated by means of a muon transportation model, which calculates all physical processes by which a muon loses kinetic energy while travelling through matter. The particle physics community has a great variety of particle simulators, the most prominent being GEANT4 (Agostinelli et al., 2003), a Monte Carlo based simulator. These have the advantage that stochastic processes resulting in energy loss are simulated according to their probabilistic

occurrence - an upside that has to be traded off for longer computation times. In contrast to obtaining the full energy loss distribution, lightweight alternatives often resort to the calculation of only the mean energy loss. The solution of the resulting differential equation can even be tabulated, as has been done by Groom et al. (2001). The interplay of these four submodules allows for the simulation of muon fluxes at the detector sites that are mostly located in an underground environment.

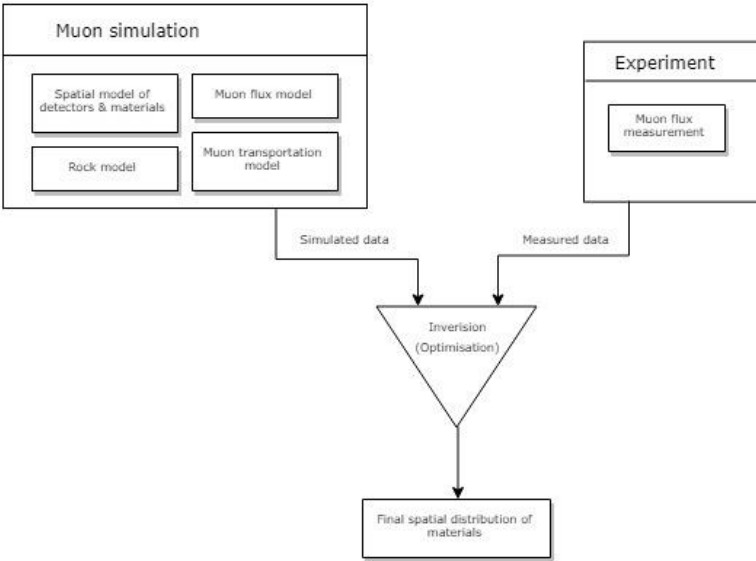

**Figure 2: A schematic flowchart showing the different involved models in a muon tomographic experiment. The muon simulation consists of a model for rocks, detectors, the cosmic ray flux and a particle physical model on how muons lose energy upon travelling through rocks. These models allow for a synthetic data set to be computed, which will be compared to the actual measured data from the experiment. An optimisation problem then solves for the best set of parameters.**




## 1.2 The need for a consistent inversion environment

The sole combination of the aforementioned four submodules does not fully justify the need for a new software, as cosmic ray flux models as well as rock models can also be programmed within existing Monte Carlo simulators such as GEANT4 (Agostinelli et al., 2003) or MUSIC (Kudryavtsev, 2009). Unfortunately, the application of such a Monte Carlo approach requires a rather good understanding of programming and nuclear physics processes. Thus, it might prove time-consuming to programme a specific code. Moreover, these codes are often written in a specialised programming language such as C++.

Third, the compatibility between different modules (e.g. cosmic ray flux and energy loss) may be severely hampered, if the programme interfaces are not taken into consideration. It might be even worse if the two modules are written in two different programming languages. In addition, one has to carefully evaluate the benefit of such an undertaking, especially if the resulting code will most likely be tailored only to a specific problem. We thus see the need for a versatile, user-friendly simulator, which allows users not only to quickly perform the necessary calculations without the need of additional coding,

but also tailor the individual models to custom needs. A new simulator can be more useful if an inversion functionality is already included. As can be seen in Fig. 2, the inversion compares the simulated flux data with the measured ones. It also attempts to reduce the discrepancy between measurements and simulations by optimising the parameters in the simulation, namely material density and the thickness distribution of the overlying materials. This results in a density- or structural rock model, which best reproduces the measured data. As the mathematical optimization in muon tomography generally is

nonlinear, one has to employ nonlinear solvers or even Monte Carlo techniques. This circumstance encourages us to work with a lightweight version of a muon transport simulator, because a nonlinear inversion of Monte Carlo simulations, although mathematically preferable, is computationally prohibitive. This allows us to make use of methods from the Bayesian realm, that thrive when measurements from different sources have to be combined into a single comprehensive model. With the code presented in this paper, we aspire to make muon tomography accessible to a broader geoscientific

community, as the know-how in this field is mainly concentrated in particle physics laboratories. We want to provide the tools for Earth scientists, or users that are mainly focused on the application of the method, so that they can perform their own analyses.

In this contribution we present our new code, SMAUG (Simulation for Muons and their Applications UnderGround), that allows a broader scientific community to plan and analyse muon tomographic experiments more easily, by providing them

with data analysis and inversion tools. Specifically, we describe the governing equations of the physical models, and the mathematical techniques that were used. Chapter 2 depicts how the muon flux simulation is conducted by its submodules and how a muon flux simulation is performed. Chapter 3 then dives into the inversion module and explains how the parameters of the inferred density/rock-model can be estimated based on measured data. This chapter includes a description of the model and data errors and an explanation on how a subsurface material boundary can be constructed. Chapter 4

provides a short overview of the program, explaining which functionality can be found in which source code. Chapter 5 discusses the model's performance based on the data that we collected in the framework of an earlier experimental campaign





(see supplement of Nishiyama et al., 2017). Chapter 6 then concludes this study by outlining a way of how this code can be developed further to fit the needs of the muon tomography and geology community.

## 2 The forward model: Muon flux simulation

In geophysical communities this part is generally known as the forward model, i.e. a mathematical model which calculates synthetic data for given "model" parameters. In muon tomography experiments this forward model consists of different physical models which are serially connected.

### 2.1 Cosmic ray flux model

The nature of the data used in muon tomography generally consists of several counts within a directional bin, defined by two
polar and two azimuthal angles. Additionally, the measurement is taken over a defined period of time, as well as over a given extent within the detector area. The simulated number of muons, in the i-th bin, can be calculated by this integral,

$$N_{\mu,i}^{sim} = \int_T \iint_\Omega \iint_A \int_E \frac{dI}{dE} \, dE \, dA \, d\Omega \, dT \, . \tag{1}$$

Here, T denotes the exposure time interval, A the detector area, $\Omega$ the solid angle of the bin and E the energy range of the muons that were able to be registered by the detector. There are various differential muon flux models, also referred to as the
integrand in Eq. (1), that can be employed at this stage. Lesparre et al. (2010) provide a good overview on the different flux models, which can broadly be divided in two classes. On the one hand there are theoretical models, which capture the manifold production paths of muons and condense them in an analytical equation, e.g. the Tang et al., (2006) model. They contrast, on the other hand, with empirical models that were generated by fitting formulae to the results of muon flux measurements. The model of Bugaev et al. (1998) falls into this category, with later adjustments for different zenith angles
(Reyna, 2006) and altitude (Nishiyama et al., 2017), which are also utilised in this study. The details of the formula are explained in Appendix A. The evaluation of Eq. (1) is rather cumbersome as strictly speaking several of the integration variables depend on each other. We may facilitate the calculation by considering that the differential muon flux model is only dependent on energy, E, and zenith angle, $\theta$ whereas the effective area, $\Delta A_{eff,i}$, is solely dependent on the orientation of the bin. This is the case because muons do not necessarily hit the detector perpendicularly, such that the effective target
area is usually smaller. By averaging over the zenith angle and keeping the bin size reasonably small, we may approximate Eq. (1) by

$$N_{\mu,i}^{sim}(E_{cut,i}) = \int_{E_{cut,i}}^{\infty} \frac{dI}{dE}\left(E, \widehat{\theta_\iota}\right) dE * \Delta T * \Delta A_{eff,i}(\widehat{\varphi_\iota}, \widehat{\theta_\iota}) * \Delta\Omega_i \, , \tag{2}$$

where $\Delta T$ is the exposure time and $\Delta A_{eff,i}$ is the effective detector area. $\Delta\Omega_i$ is the solid angle, $\widehat{\varphi_\iota}$ and $\widehat{\theta_\iota}$ are the mean azimuth and zenith angle of the i-th bin, respectively. $E_{cut,i}$ describes the energy needed for a muon to enter the





detector. $\Delta A_{eff,i}$ has to be scaled by the cosine of the angle between the bin direction and the detector facing direction, which can be calculated using the formula for a scalar product,

$$\Delta A_{eff,i} = \Delta A * \frac{\vec{n} \cdot \vec{d}_\mu(\hat{\varphi}_i, \hat{\theta}_i)}{\|\vec{n}\| \|\vec{d}_\mu(\hat{\varphi}_i, \hat{\theta}_i)\|}, \tag{3}$$

where $\vec{n}$ is the normal vector to the detector surface and $\vec{d}_\mu(\hat{\varphi}_i, \hat{\theta}_i)$ is the mean vector of muon incidence within the i-th bin, both of which can be chosen to feature unit length. Evaluating the scalar product in spherical coordinates, Eq. (3) yields

$$\Delta A_{eff,i} = \Delta A * \left[ \sin(\theta_d) \sin(\hat{\theta}_i) \cos(\varphi_d - \hat{\varphi}_i) + \cos(\theta_d) \cos(\hat{\theta}_i) \right]. \tag{4}$$

Here, $\theta_d$ and $\varphi_d$ are the zenith and azimuth angles of the detector facing direction. It is important to note that except for $E_{cut,i}$ all variables in Eq. (2) are predetermined by the experimental setup ($\Delta T$, $\Delta A$) as well as by the data processing ($\hat{\varphi}_i$, $\hat{\theta}_i$), such that the number of muons $N_{\mu,i}^{sim}$ can be interpreted as a function of one variable, $E_{cut,i}$ only.

## 2.2 Muon transportation model

Since muons permanently lose energy when travelling through matter, they also need a certain amount of energy to enter the detector. This energy, $E_{cut,i}$, was introduced in Eq. (2) and is called the cutoff energy. If the detector is now positioned underground, the muons have to traverse more matter to reach the detector and consequently need a higher initial energy to reach the target. For this purpose, we introduce the new variable $E_0$, which refers to the energy needed to penetrate the detector, and we reinterpret $E_{cut,i}$ as the minimum energy that is required to traverse the matter and to be registered at the

detector. For the goal of studying the interactions between particles and matter, physicists regularly use energy loss models. We base our calculations in large parts on the equations of Groom et al. (2001), where the energy loss of a muon along its path is described by an ordinary differential equation of 1st order,

$$-\frac{dE}{dx} = \rho(x) * [a(x, E) + E * b(x, E)]. \tag{5}$$

In Eq. (5), $\rho$ denotes the density of the traversed material, and $a$ and $b$ are the ionisation loss and radiation loss parameters

respectively. The radiation loss parameter groups the effects related to the bremsstrahlung, $b_{brems}$, the pair-production, $b_{pair}$, and the photonuclear interactions, $b_{photo}$, where

$$b(x, E) = b_{brems}(x, E) + b_{pair}(x, E) + b_{photo}(x, E). \tag{6}$$

Each of the radiative process is, in turn, calculated through

$$b_k = \frac{N_A}{A} \int_0^1 v \frac{d\sigma_k}{dv} dv, \tag{7}$$

where $k \in \tilde{K} = \{\text{bremsstrahlung, pair-production, photonuclear}\}$ is the set of radiative processes, $N_A$ is Avogadro's number, $A$ is the atomic weight of the traversed material, $v$ is the fractional energy transfer, and $d\sigma_k/dv$ the differential cross-section




of the process. Eq. (7) becomes important when modelling errors have to be included (see Ch. 3). For a detailed discussion of the equations for $a$ and $b$ we refer to Groom et al. (2001). The only exception in Eq. (6) is $b_{pair}$, which is calculated after GEANT4 (Agostinelli et al., 2003). We selected the solution of these latter authors because it is computationally less time

consuming. As the two results agree within 1 %, we deem it acceptable to exchange the two differential cross-sections.

Because Eq. (5) describes the energy loss in response to the interaction with a single-element material, certain modifications have to be made to make it also valid for rocks, which in this context represent a mixture of minerals and elements. In this case, the modified equation takes an equivalent form to Eq. (5) when replacing $\rho, a, b$ with their mixture counterparts $\{\rho\}_{rock}, \{a\}_{rock}, \{b\}_{rock}$ (Lechmann et al., 2018), thus yielding

$$-\frac{dE}{dx} = \{\rho(x)\}_{rock} * [\{a(x,E)\}_{rock} + E * \{b(x,E)\}_{rock}] .\tag{8}$$

We show in Appendix B3 how the rock model (explained in Ch. 2.3) can be used to determine these quantities.

By applying a change of variables to Eq. (8), i.e. $x' = -x$, the energy loss equation can be transformed to an energy gain equation. This has the advantage of being much easier to solve than the "final value problem" in Eq. (8). We can reorganise Eq. (8) into an initial value problem by setting the initial energy to $E_0$,

$$\frac{dE}{dx} = \{\rho(x)\}_{rock} * [\{a(x,E)\}_{rock} + E * \{b(x,E)\}_{rock}]\tag{9}$$

$$E(0) = E_0 .$$

In this context $E_0$ is the minimal energy needed for a muon to penetrate the detector, which can be influenced by the detector design. Equation (9) is a well-investigated problem that can be solved by numerous methods. In our work we employ a standard Runge-Kutta integration scheme (see for example Stoer and Bulirsch, 2013), with a step size of 10 cm. As a result,

it is now possible to write the cut-off energy in a functional form, where

$$E_{cut,i} = rk(\vec{x}_i, \vec{\rho}, \vec{c}) .\tag{10}$$

Here $rk(\cdot)$ is the function that returns the Runge-Kutta solution of Eq. (9) for defined thicknesses of materials, $\vec{x}_i$, with densities $\vec{\rho}$ and compositional parameters $\vec{c}$. Thickness and density are allowed to be vectors, as there may be more than just one material. In this case, the final energy, after the muon has passed through the first segment of materials, is the initial

energy for the second segment, etc. In order to speed up the computations – especially the calculation of the pair production cross-section, which includes two nested integrations. In particular, a log-log table of muon energy vs. radiation loss parameters is produced, from which the b-values, see Eq. (7), can be interpolated. We justify this approach because the radiative losses are almost linear in a log-log plot, as can be seen in Fig. 33.1 of Tanabashi et al. (2018, p.447) for the example of copper. The general shape of the energy loss function remains the same for various materials even if the absolute

values differ.





## 2.3 Rock model

Equation (10) shows that for the calculation of the cut-off energy two types of material parameters are required, which are the material density $\vec{\rho}$ and its average composition $\vec{c}$. The pre-tabulated values from Groom et al. (2001), however, include only pure elements as well as certain compounds. To extract the relevant parameters in a geological setting, a realistic rock

model is needed. In an earlier work (Lechmann et al., 2018) we have shown how an integrated rock model can be constructed and how the physical parameters for a realistic rock can be retrieved. In the present work we generally use the same approach, apart from a few aspects. First, we measured the average material density directly in the laboratory, using various techniques which are explained in detail in Appendix B1. Second, in order to be able to compare the results of this study with the ones in the Nishiyama et al. (2017) publication, we consider a rock composition that corresponds to a density-

modified standard rock. This is applicable, as the rock in the study of Nishiyama et al. (2017) is mostly of granitic/gneissic origin, with thicknesses rarely larger than 200 m, with the consequence that the differences are negligible. However, as the inclusion of compositional data is a planned feature for a future version of our code, we decided to include the theoretical treatment in this work. Hence, all equations are tailored to include the statistical description of such data. Compositional data for whole rock samples, which can be scaled to outcrop scale, are usually presented in one of two forms, the first being the

measurement results of X-Ray Diffraction (XRD). This kind of data yields the mineral phases within a rock. Unfortunately, XRD is a rather time-consuming method. This is the reason why in muon tomographic experiments researchers often resort to a bulk chemical analysis of the rock, which is the second form of compositional data. This type of data is usually the output of dedicated X-Ray Fluorescence (XRF) measurements, describing the bulk rock composition by major oxide fractions. We note here that by the absence of information on the spatial distribution of mineral phases within a rock, we

implicitly infer a homogeneous mixture of elements within the rock itself, which is thus different from our previous work (Lechmann et al., 2018). From a particle physics perspective this does not pose a real problem as the difference to a mixture of minerals is rather small. Nevertheless, we lose the power to obtain meaningful inferences that could be drawn if compositional information is being considered. As the present work aims to infer positions and uses material parameters as constraints, we can accept this drawback. Details on how compositional parameters are derived from XRF measurements,

including an example, can be found in Appendix B2, and an explanation of the related influence on the energy loss equation in Appendix B3.

## 2.4 Spatial models of detectors and materials

In addition to the above explained physical models, we may also utilise available spatial data for our purposes. In this context, the use of a digital elevation model (DEM) of the surface allows the visualisation of the position of the detectors

relative to the surface, as well as the spatial extent of the bins. Additionally, it allows us to determine the location where these bins intersect with the topographic surface. As a first deliverable, we can draw conclusions on which bins consist of how many parameters. For example, if we know that the detector is located underground and that there is ice at the surface,



we can already infer the existence of at least 2 materials (rock and ice). For this purpose, we wrote the script "modelbuilder.py", which allows the user to attach geographic and physical information to the selected bins. This process of
building a coherent geophysical model is needed for the subsequent employment of the inversion algorithm to process all the data.

## 3 The inverse model: A Bayesian perspective

As stated in the Introduction, we solve the inversion by using Bayesian methods. This needs an explanation as to why we chose this way and not another. First, the equations in Ch. 2 enable us to calculate a synthetic dataset for fixed parameter
values. There, one can see that the governing equations constitute a nonlinear relationship between parameter values and measured data. Despite this being of no particular interest in the forward model, the estimation of the parameters from measured data is rendered more complicated. Among muon tomographers, linearised versions have been extensively used with deterministic approaches (e.g. Nishiyama et al., 2014; Rosas-Carbajal et al., 2017), which are successfully applicable when the density or the intersection boundaries are the only variables. When deterministic approaches are viable, they
efficiently produce good results. Descent algorithms or, generally speaking, locally optimising algorithms, offer a valid alternative, as they could cope with the nonlinearity of the forward model, while including all desired parameters. Even though these algorithms suffer from possible non-uniqueness solutions (i.e. the solution depends heavily on the starting model, possibly yielding multiple solutions), the main problem is the calculation of the derivatives of the forward model with respect to the parameter values. The analytical calculation of the derivatives is enormously tedious because the cut-off
energy results from a numerical solver of a differential equation, as can be seen in Eq. (9) & (10). Unfortunately, numerical derivatives do not produce better results, because they might easily produce artefacts, which are hard to track down. This is especially true if the derivative has to be taken from a numerical result, which is always slightly noisy. In that case the differentiation amplifies the "noise", resulting in unreliable gradient estimates. A good overview over deterministic inversion methods can be found in Tarantola (2005).
The reasons stated above and our goal to include as much information on the parameters as possible nudges us towards employing probabilistic methods. Those approaches are also known as Bayesian methods. The main feature that distinguishes them from the deterministic methods described above is the consistent formulation of the equations and additional information in a probabilistic way, i.e. as probability density functions (pdf). This allows us to (i) incorporate, for example, density values that were measured in the lab (including its error), (ii) set bounds on the location of the material
interface, or (iii) define a plausible range for the composition of the rock. All these changes act on the pdf of the respective parameter and naturally integrate into the Bayesian inversion. Readers may find the book of Tarantola (2005) very resourceful for the explanation and illustration of probabilistic inversion. Several studies in the muon tomography community have already employed such methods with success (e.g. Lesparre et al., 2012; Barnoud et al., 2019).





The flexibility of being able to include as much information on the parameters as we consider useful comes at the price of

having to solve the inversion in a probabilistic way. This can either be done using Bayes' Theorem and solving for the pdfs

of the parameters of interest, or if the analytical way is not possible by employing Monte Carlo techniques. As the presence

of a numerical solver renders the analytical solution impossible, we resort to the Monte Carlo approaches. In the following

sections we guide the reader through the various stages of how such a probabilistic model can be set up, how probabilities

may be assigned, and how the inversion can finally be solved.

## 3.1 Probabilistic formulation of the forward model

The starting point for a probabilistic formulation is denoted by the equations that were elaborated in Ch. 2. These

deterministic equations need to be upgraded into a probabilistic framework, where their attributed model and/or parameter

uncertainties are inherently described. In the following paragraphs we describe how each model component can be expressed

by a pdf before the entire model is composed at the end of this subchapter. The model is best visualised by a directed acyclic

graph (DAG), i.e. see Kjaerulff and Madsen (2008), that depicts which variables enter the calculation at what point. For our

muon tomography experiment this is visualised in Fig. 3. In the following the pdfs are denoted with the bold Greek letter $\boldsymbol{\pi}$

to differentiate them from normal parameters.

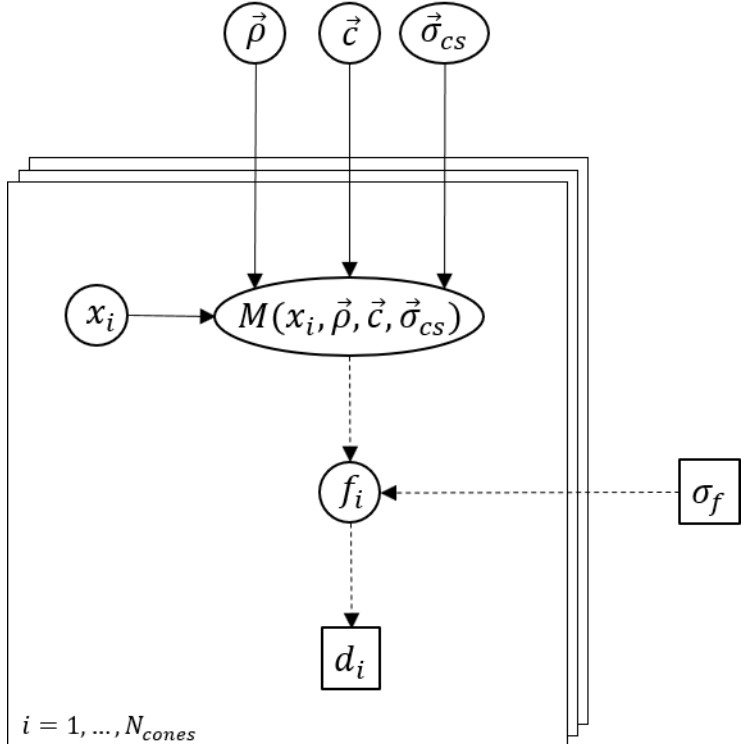

**Figure 3: Directed acyclic graph (DAG) for the problem of muon tomography. Variables in a square (□) denote fixed, i.e. known**
**values and variables in a circle/ellipse (○) are generally unknown and have to be represented by a pdf. Solid arrows (→) denote a**
**deterministic relation, i.e. within a physical model, whereas dashed arrows (⇢) indicate a probabilistic relationship, i.e. a**





parameter within the statistical description of the variable. $\vec{\rho}, \vec{c}$ are the density and composition for different materials, whereas $\vec{\sigma}_{cs}$ contains the errors on the physical cross-sections in the energy-loss equation. $\sigma_f$ describes the error on the cosmic-ray flux model. Within each cone, $x_i$ is the position of the bedrock-ice interface, $M(x_i, \vec{\rho}, \vec{c}, \vec{\sigma}_{cs})$ is the calculated flux (i.e. energy-loss model
and flux model combined), $f_i$ the actual muon flux and $d_i$ the observed number of muon tracks.

### 3.1.1 Muon data

The data in muon tomography experiments are usually count data, i.e. a certain number of measured tracks within a directional bin, which has been collected over a certain exposure time and detector area. As the measured number of muons is always an integer, we may model such data by a Poisson distribution,

$\boldsymbol{\pi}(d_i|N_i) = \frac{N_i^{d_i} e^{-N_i}}{d_i!},$ (11)

where $d_i$ denotes the measured number of muons in the i-th bin and $N_i$ is the poisson parameter in the same bin, which can be interpreted as mean and variance of this distribution. Equation (11) may be rewritten in terms of a flux, $f_i$ by

$\boldsymbol{\pi}(d_i|f_i) = \frac{(f_i \Delta Ex_i)^{d_i} e^{-(f_i \Delta Ex_i)}}{d_i!},$ (12)

where $f_i$ is the muon flux in the i-th bin and

$\Delta Ex_i = \Delta A_{eff,i} * \Delta T * \Delta \Omega_i$ (13)

is the exposure, in which $\Delta A_{eff,i}$ is the effective total detector area from Eq. (4), $\Delta T$ is the exposure time and $\Delta \Omega_i$ is the solid angle.

### 3.1.2 Flux model

The next step is to set up a probabilistic model for the muon flux. First, we observe that "flux" is a purely positive parameter,
i.e. $f_i \in [0, \infty)$. Thus, it is natural to model it by a lognormal probability distribution if estimates of mean and variance are readily available. The uncertainty on the muon flux is generally taken around 15% of the mean value. As it is possible, by Eq. (2), to calculate a flux for a given cut-off energy, which we interpret as the mean of the non-logarithmic values, the parameters of the lognormal distribution (i.e. $\mu_{f_i}, \sigma_{f_i}^2$) may be expressed by

$\sigma_{f_i}^2 = \ln\left(1 + \left(\frac{F_i(E_{cut,i})*0.15}{F_i(E_{cut,i})}\right)^2\right) = \ln(1.0225)$ (14)

and

$\mu_{f_i} = \ln\left(F_i(E_{cut,i})\right) - \frac{\sigma_{f_i}^2}{2},$ (15)

which yield the probability density function for the flux, conditional on the cut-off energy





$$\pi\left(f_i \mid \mu_{f_i}, \sigma_{f_i}^2\right) = \pi\left(f_i \mid E_{cut,i}\right) = \frac{1}{\sqrt{2\pi} * f_i * \sigma_{f_i}} \exp\left(-\frac{1}{2}\left(\frac{\ln(f_i) - \mu_{f_i}}{\sigma_{f_i}}\right)^2\right). \tag{16}$$

### 3.1.3 Energy loss model

The energy loss model has multiple sources of errors that have to be taken into account. Most notably, the relative errors on the different physical cross-sections are given by Groom et al. (2001) as $\varepsilon_{ion} = 6\,\%, \varepsilon_{brems} = 1\,\%, \varepsilon_{pair} = 5\,\%, \varepsilon_{photonucl} = 30\,\%$ . As it is not clearly stated as to what this error relates to, i.e. one or more standard deviations, we interpret an error like $\varepsilon_{ion} = 6\,\%$ as: "within a factor of 1.06", which can be written as

$$\frac{\sigma_k}{(1 + \varepsilon_k)} \leq \sigma_k \leq \sigma_k(1 + \varepsilon_k), \tag{17}$$

where $k \in K = \{$ionisation, bremsstrahlung, pair-production, photonuclear$\}$. Dividing this inequality by $\sigma_k$ and taking the logarithm yields

$$-\ln(1 + \varepsilon_k) \leq 0 \leq \ln(1 + \varepsilon_k). \tag{18}$$

Thus, we may attribute a Gaussian pdf in the log-space for a "log-correction factor, $l\sigma_k$" by setting its mean to zero and its standard deviation to $\ln(1 + \varepsilon_k)$, i.e.

$$\pi(l\sigma_k) = \frac{1}{\sqrt{2\pi} * \ln(1 + \varepsilon_k)} \exp\left(-\frac{1}{2}\left(\frac{l\sigma_k}{\ln(1 + \varepsilon_k)}\right)^2\right). \tag{19}$$

With a change of variables, using the Jacobian rule as explained in Tarantola (2005), we get

$$\pi(\sigma_k) = \frac{1}{\sqrt{2\pi} * \sigma_k * \ln(1 + \varepsilon_k)} \exp\left(-\frac{1}{2}\left(\frac{\ln(\sigma_k)}{\ln(1 + \varepsilon_k)}\right)^2\right), \tag{20}$$

the lognormal pdf for the correction factor. The pdf for the cross-section uncertainty $\pi(\vec{\sigma}_{cs})$ can now be written as a product of the four different pdfs described by Eq. (20)

$$\pi(\vec{\sigma}_{cs}) = \prod_{k \in K} \pi(\sigma_k), \tag{21}$$

as the errors of the physical cross-sections are stochastically independent from each other.

The calculated energy loss depends also on the material parameters and subsequently on their uncertainties. However, these will be explained in detail in Ch. 3.1.4. A last error enters by the numerical solution of the ordinary differential equation, Eq. (9). We decided not to model this error, as its magnitude is directly controlled by the user (by setting a small enough step

length in the Runge-Kutta algorithm) and thus can be made arbitrarily small. Lastly, we assume that all the errors in the energy loss model are explained by uncertainties in the cross sections as well as in the material parameters. Although this assumption is rather strong, since it excludes the possibility of a wrong model, we argue that this approach works as long as





the variation in these parameters can explain the variation in the calculated cut-off energy. If this requirement is met, we may model the pdf for the energy loss model as a delta function,

$$\pi\left(E_{cut,i}|\vec{\sigma}_{cs}, \vec{\rho}, \vec{c}, x_i\right) = \delta\left(E_{cut,i} - rk(\vec{\sigma}_{cs}, \vec{\rho}, \vec{c}, x_i)\right), \tag{22}$$

where $\vec{\sigma}_{cs} = (\sigma_k)$, $\vec{\rho}$ is the vector of all material densities, $\vec{c}$ is the vector of all compositions and $x_i$ is the vector of thicknesses of segments used in this cone. It is now already possible to eliminate $E_{cut,i}$ as a parameter by first multiplying Eqs. (16) & (22), which yields

$$\pi(f_i, E_{cut,i}|\vec{\sigma}_{cs}, \vec{\rho}, \vec{c}, x_i) = \pi\left(f_i|E_{cut,i}\right) * \pi\left(E_{cut,i}|\vec{\sigma}_{cs}, \vec{\rho}, \vec{c}, x_i\right). \tag{23}$$

From this expression it is possible to marginalise the parameter $E_{cut,i}$, by simply integrating over it, i.e.

$$\pi(f_i|\vec{\sigma}_{cs}, \vec{\rho}, \vec{c}, x_i) = \int \pi(f_i, E_{cut,i}|\vec{\sigma}_{cs}, \vec{\rho}, \vec{c}, x_i)\, dE_{cut,i}. \tag{24}$$

Due to the presence of the delta function in Eq. (22), this integral is solved analytically resulting in

$$\pi(f_i|\vec{\sigma}_{cs}, \vec{\rho}_i, \vec{c}_i, x_i) = \frac{1}{\sqrt{2\pi} * f_i * \sigma_{f_i}} \exp\left(-\frac{1}{2}\left(\frac{\ln(f_i) - \mu_{f_i}}{\sigma_{f_i}}\right)^2\right), \tag{25}$$

where the parameters are given by

$$\sigma_{f_i}^2 = \ln\left(1 + \left(\frac{M(\vec{\sigma}_{cs}, \vec{\rho}, \vec{c}, x_i) * 0.15}{M(\vec{\sigma}_{cs}, \vec{\rho}, \vec{c}, x_i)}\right)^2\right) = \ln(1.0225) \tag{26}$$

and

$$\mu_{f_i} = \ln(M(\vec{\sigma}_{cs}, \vec{\rho}, \vec{c}, x_i)) - \frac{\sigma_{f_i}^2}{2}. \tag{27}$$

Please note that $M(\vec{\sigma}_{cs}, \vec{\rho}, \vec{c}, x_i) = F_i(rk(\vec{\sigma}_{cs}, \vec{\rho}, \vec{c}, x_i))$ describes the combined parts of the forward model that include the energy loss and the integrated flux calculation, which is basically a composition of functions.

### 3.1.4 Rock model

The density model can take different forms of probability densities (see Appendix B1), such as normal, lognormal, uniform, etc. For either form, it is possible to describe it by a generic function $\pi(\vec{\rho})$, which is a short version for a multidimensional pdf, i.e. $\pi(\rho_{ice}, \rho_{rock})$ if the i-th cone is known to consist of two segments with two specific densities. Equivalently, the pdf for the composition (see Appendix B2) is either fixed or a multidimensional Gaussian distribution in the space of log-ratios. Thus $\pi(\vec{c})$ can be split up to $\pi(c_{ice}, c_{rock})$, like in the example above. Generally, we may assume that in our problem j different materials exist. We note here that the description of the composition is already probabilistic. However, the inversion in that case is not functional and works only with the mean values of the multidimensional Gaussian. The support for composition inversion is planned for a future version of the code.





The situation for the thicknesses of the segments, $\pi(x_i)$, within the i-th cone presents itself in a similar way as for the
compositions (e.g., Fig. 1). As the total material thickness is known (detector position and digital elevation models are given), the sub-space containing the thickness parameter is endowed with the same mathematical structure as the one containing the composition parameter (i.e. one sum constraint), if the cone consists of more than just one segment. One can therefore safely assume that the thickness parameters live in a log-ratio space, within which we a-priori possess no information about the parameters. Thus, we attribute the thickness parameters a multidimensional uniform distribution
within the log-ratio space.

### 3.1.5 The Joint probability density function

With the help of the DAG, introduced in Fig. 3, it is now straightforward to factorise the joint probability distribution for the whole problem, as their structure is equal. This results in

$$\pi(\vec{d}, \vec{f}, \vec{\sigma}_{cs}, \vec{\rho}, \vec{c}, \vec{x}) = \prod_{i=1}^{N_{cones}} \pi(d_i|f_i)\pi(f_i|\vec{\sigma}_{cs}, \vec{\rho}, \vec{c}, x_i)\pi(x_i) * \prod_{j=1}^{N_{materials}} \pi(\rho_j)\pi(c_j) * \prod_{k\in K} \pi(\sigma_k), \tag{28}$$

or equivalently (and this will also be of a much better use later on) the log joint pdf

$$l\pi(\vec{d}, \vec{f}, \vec{\sigma}_{cs}, \vec{\rho}, \vec{c}, \vec{x}) = \sum_{i=1}^{N_{cones}} l\pi(d_i|f_i) + l\pi(f_i|\vec{\sigma}_{cs}, \vec{\rho}, \vec{c}, x_i) + l\pi(x_i) + \sum_{j=1}^{N_{materials}} l\pi(\rho_j) + l\pi(c_j) + \sum_{k\in K} l\pi(\sigma_k), \tag{29}$$

where the prefix "$l$" denotes the logarithm of the pdf. This has the benefit of reducing the size of numbers that the code has to cope with. Moreover, many computational statistics packages already have this feature included, which renders it easy to use.

Equation (28) depicts the full joint pdf. However, the relations between the parameters, as shown by the DAG (see Fig. 3), classify this model as a hierarchical model (Betancourt and Girolami, 2013). The key characteristic of such models is their tree-like parameter structure, i.e. the measured number of muons is related to the thickness or the density of the material by the flux parameter only, which "relays" the information. A central problem of such models is the presence of a hierarchical "funnel" (see Fig. 2 & 3 of Betancourt and Girolami, 2013), which renders it very difficult for standard MCMC methods to
adequately sample the model space. In high-dimensional parameter spaces this problem exacerbates even more.

Our aim to provide a simple and easy-to-use program somewhat contradicts this necessity of a sophisticated method (which inevitably requires the user to possess a strong statistical background). As the main problem is the rising number of parameters, it should be possible to mend the joint pdf by imposing thought-out simplifications.

We first get rid of the flux parameter, as for our problem it merely is a nuisance parameter. This is an official term for a
parameter in the inversion which is of no particular interest but still has to be accounted for. Here specifically, we integrate over all possible values of the muon flux, $\vec{f}$ within its uncertainty, so that we can relate the results of the energy loss calculation (encoded in $\mu_{f_i}$; see Eq. 27) directly to the measured number of muons, $d_i$. This effectively reduces the number





of parameters and thus the number of dimensions of the model space. This can be achieved by marginalising the flux parameter out of the joint pdf,

$$\boldsymbol{\pi}\big(\vec{d}, \vec{\sigma}_{cs}, \vec{\rho}, \vec{c}, \vec{x}\big) = \int \boldsymbol{\pi}\big(\vec{d}, \vec{f}, \vec{\sigma}_{cs}, \vec{\rho}, \vec{c}, \vec{x}\big)\, d\vec{f}\,. \tag{30}$$

This effectively reduces to a problem where the new likelihoods have to be calculated (as $d_i$ is given)

$$\boldsymbol{\pi}(d_i|\vec{\sigma}_{cs}, \vec{\rho}, \vec{c}, x_i) = \int \boldsymbol{\pi}(d_i|f_i)\boldsymbol{\pi}(f_i|\vec{\sigma}_{cs}, \vec{\rho}, \vec{c}, x_i)\, df_i, \tag{31}$$

or fully,

$$\boldsymbol{\pi}(d_i|\vec{\sigma}_{cs}, \vec{\rho}, \vec{c}, x_i) = \int_0^\infty \frac{(f_i \Delta E x_i)^{d_i} e^{-(f_i \Delta E x_i)}}{d_i!} \frac{1}{\sqrt{2\pi} * f_i * \sigma_{f_i}} \exp\left(-\frac{1}{2}\left(\frac{\ln(f_i) - \mu_{f_i}}{\sigma_{f_i}}\right)^2\right) df_i\,, \tag{32}$$

where $\mu_{f_i}$ and $\sigma_{f_i}$ are given by Eqs. (26) and (27), respectively. This integral is not solvable analytically but can be evaluated by numerical integration schemes. The likelihood has a maximum when the Poisson and the log-normal pdfs fully overlap. Interestingly, this directly shows the trade-off between the flux model and the data uncertainty. Usually, we want to measure enough muons so that the statistical counting error is smaller than the systematic uncertainty of the flux model (i.e. the width of the Poisson pdf is smaller than the width of the log-normal pdf). This can be controlled directly by the exposure of the experiment, via a larger detector area, a coarser binning, or a longer exposure time.

This marginalisation roughly halves the number of parameters, but there is still another simplification, which we may use. Many muon tomography applications deal with a two-material problem, while there may also be measurement directions where only one material is present. If we conceptually split those two problems and solve them independently, it is possible to further reduce the number of simultaneously modelled parameters. In the study of Nishyiama et al. (2017), the results of which we will use later, these two cases encompass bins where we measured only rock and others where we know there is ice and rock. The joint pdf for rock bins subsequently is

$$\boldsymbol{\pi}\big(\vec{d}, \vec{\sigma}_{cs}, \rho_{rock}, c_{\text{rock}}\big) = \prod_{i=1}^{N_{cones}^{rock}} \boldsymbol{\pi}(d_i|\vec{\sigma}_{cs}, \rho_{rock}, c_{\text{rock}}) * \boldsymbol{\pi}(\rho_{rock})\boldsymbol{\pi}(c_{rock}) * \prod_{k \in K} \boldsymbol{\pi}(\sigma_k), \tag{33}$$

which leaves the problem effectively with only a handful of parameters. Solving Eq. (33), for the rock density we retrieve $\widetilde{\boldsymbol{\pi}}(\rho_{rock})$, the posterior marginal pdf for the rock density. We refer the reader to Ch. 3.2 for the details of how to solve this inverse problem. Theoretically we could also retrieve $\widetilde{\boldsymbol{\pi}}(\rho_{ice})$, but this would require the detector to be positioned within the glacier (see chapter 5.2 and Fig. 1 for a description of the experimental design of the Nishyiama et al. (2017) study), which poses more of a practical difficulty than a mathematical one.

For the second problem, we can interpret $\widetilde{\boldsymbol{\pi}}(\rho_{rock})$ as the new prior pdf for the rock density. At this point we employ one last simplification by assuming that the parameters between different cones are independent form each other. This is a rather strong presumption, which must be justified. The main mathematical problem lies in consideration of the hierarchical nature of the density parameter, which is the same for each cone and therefore not independent in different cones. We, however,





argue that in cones with two materials, there are more parameters than in bins with only rock, such that we may expect the posterior pdf of the rock density of these second kind of models to be less informative than the posterior rock density pdf of Eq. (33). This, in turn, means that the posterior rock density pdf of the two-material model largely equals the prior one if we

select the posterior of the first kind of models as the prior of the second kind of models. The same is valid for the composition $\vec{c}_i$ and the cross-section error parameters $\vec{\sigma}_{cs}$. As long as this assumption is valid, we may decompose the joint pdf into independent joint pdfs for each cone

$$\pi(d_i, \vec{\sigma}_{cs,i}, \vec{\rho}_i, \vec{c}_i, x_i\,) = \pi(d_i|\vec{\sigma}_{cs,i}, \vec{\rho}_i, \vec{c}_i, x_i)\pi(x_i)\prod_{j=1}^{N_{materials}}\widetilde{\pi}(\rho_{ij})\widetilde{\pi}(c_{ij}) * \prod_{k\in K}\widetilde{\pi}(\sigma_{ik}).\tag{34}$$

Our inversion program enables the user to choose the type of model parametrisation, which is either the full hierarchical
model given by Eqs. (28) & (29), or the simplified single-cone-bin inversion model ("Sicobi"-model) given by Eqs. (33) & (34).

## 3.2 Solution to the inverse problem

Usually in Bayesian inference, the goal is to calculate the posterior pdf, given the measured data, i.e. the quantity

$$\pi(\vec{\sigma}_{cs}, \vec{\rho}, \vec{c}, \vec{x}|\vec{d}) = \frac{\pi(\vec{d},\vec{\sigma}_{cs},\vec{\rho},\vec{c},\vec{x})}{\pi(\vec{d})}.\tag{35}$$

This can be interpreted as the inferences one may draw on the parameters in a model given measured data. The denominator on the right-hand side of Eq. (35), also called the "evidence", can be rewritten as the data marginal of the posterior, i.e.

$$\pi(\vec{d}) = \int\int\int\int \pi(\vec{d},\vec{\sigma}_{cs},\vec{\rho},\vec{c},\vec{x})\, d\vec{\sigma}_{cs}\, d\vec{\rho}\, d\vec{c}\, d\vec{x}\,.\tag{36}$$

As Eq. (36) basically describes an integration over the whole model parameter space, this may become such an extensive computation (especially when the number of model parameters is large), that it cannot be solved in a meaningful time.
However, as the evidence usually is a fixed value, the left- and right-hand side of Eq. (35) are merely scaled by a scalar and are thus proportional to each other, i.e.

$$\pi(\vec{\sigma}_{cs}, \vec{\rho}, \vec{c}, \vec{x}|\vec{d}) \propto \pi(\vec{d},\vec{\sigma}_{cs},\vec{\rho},\vec{c},\vec{x}).\tag{37}$$

This is the starting point of Monte Carlo Markov Chain (MCMC) methods.

### 3.2.1 The Metropolis-Hastings algorithm

The basic MCMC algorithm, which we also use in this study, is the Metropolis-Hastings (MH) algorithm (Hastings, 1970; Metropolis et al., 1953), which allows for the sampling of the joint pdf to obtain a quantitative sample. We note, however, that many different MCMC algorithms exist for various purposes and that the MH has no special status except for being comparatively simple to use and implement. An example of another MCMC algorithm in muon tomography can be found in Lesparre et al. (2017). The authors used a simulated annealing technique on the posterior pdf in order to extract the





maximum a posteriori (MAP) model. As every simulated annealing algorithm has some type of MH-algorithm at its core, we directly use the MH-algorithm in its original form such that we not only retrieve a point estimate but a pdf for the posterior parameter distribution. The algorithm is explained in detail by Gelman (2014), such that we only provide a short pseudo-code description.

**Algorithm 1 (Metropolis-Hastings):**

(1) Draw a starting model, $\vec{m}_0 = (\vec{\sigma}_{cs,0}, \vec{\rho}_0, \vec{c}_0, \vec{x}_0)$, by drawing $\vec{\sigma}_{cs,0}, \vec{\rho}_0, \vec{c}_0, \vec{x}_0$ from their respective prior pdfs and determine the log-pdf value of this model

    (2) Until convergence:

        a. Propose a new model according to $\vec{m}_{new} = \vec{m}_0 + J(0, c^2\Sigma)$, where $\Sigma$ is the matrix of prior variances and $c = 2.4/\sqrt{D}$ and $D$ is the number of parameters.

480         b. Evaluate log-pdf value of $\vec{m}_{new}$ and calculate:

           $r = \exp(lp(\vec{m}_{new}) - lp(\vec{m}_0))$

        c. Evaluate the acceptance probability, $p_A = \min(1, r)$ and draw a number q from the uniform distribution U(0,1).

        d. If $q < p_A$: sample $\vec{m}_{new}$ & set $\vec{m}_{new} \rightarrow \vec{m}_0$,

485            Else: sample $\vec{m}_0$

The advantage of this algorithm, compared to a "normal" sampling, lies in its efficiency. It is often not possible, or even reasonable, to probe the whole model space, as the largest part of the model space is "empty", where the pdf-value of the posterior is uninterestingly small. The fact that regions of high probability are scarce, and this becomes worse in high

dimensional model spaces, is known as the "curse of dimensionality" (Bellman, 2016). MCMC algorithms (including the here presented MH-algorithm) allows one to focus on regions of high probability, and therefore we are able to construct a reliable and representative sample of the posterior pdf. We again refer to Gelman (2014) for a discussion of why the MH-algorithm converges to the correct distribution and why we may use samples that were gained this way to estimate the posterior probability density.

**3.2.2 Assessing convergence, mixing, and retrieving the samples**

The above stated advantages, however, come at a price. First and foremost, we must ensure that the algorithm advances fast enough, but not too fast, through the model space. This is mainly controlled by the proposal distribution $J(0, c^2\Sigma)$, which is taken to be a multivariate Gaussian distribution. Ideally, the covariance matrix of the proposal distribution $\Sigma$ is equal to the covariance structure of the posterior pdf. We acknowledge that at the start of the algorithm one generally has no idea what

this looks like, but we assume that a combination of the prior variances is a reasonable starting point. After a certain number

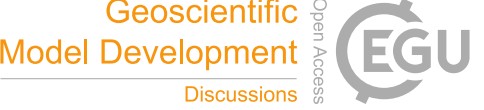



of steps, it is possible to approximate the covariance matrix of the proposal distribution with the samples taken up to at this point.

A second crucial point is the presence of a warm-up period. The starting point, which usually lies in a region of high prior probability, does not necessarily lie in a region of high posterior probability. The time it takes to move from the latter to the
former is exactly this warm-up. This can usually be visualised by a trace plot, e.g. Fig. 4, in which the value of a parameter is plotted against the number of iterations. After this warm-up phase, the algorithm can be run in operational mode and "true" samples can be collected.

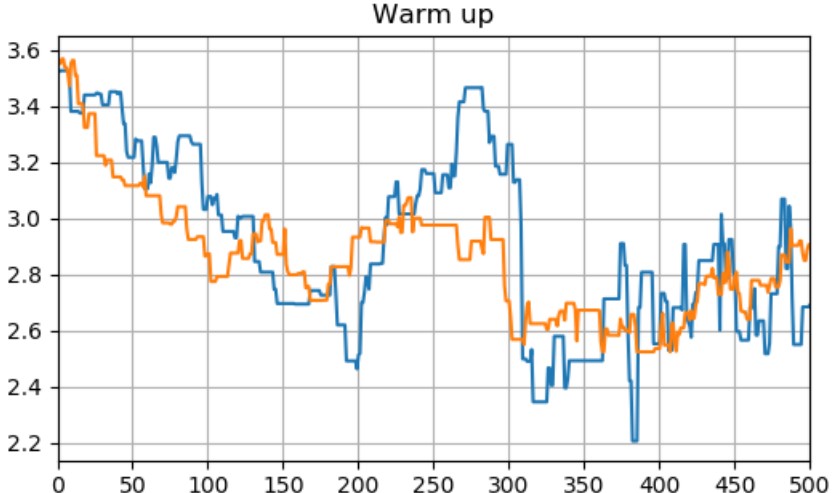

**Figure 4: Example of a trace plot (2 independent chains; blue and orange) of a MH run with 500 draws. This plot shows the**
**parameter value (y-axis; here material density in $g\ cm^{-3}$) vs. # of steps (x-axis) of a collection of cones in which we (Nishyiama et al., 2017) knew that only rock is present. This is a calculation that is included in the code base. The warm-up phase of this MH algorithm takes roughly 150 simulations indicated by the subsequent oscillation around a parameter value of $\sim 2.7 - 2.8\ g\ cm^{-3}$.**

As in a Markov Chain the actual sample is dependent on the last one, we need a criterion to argue that the samples created in that way really represent "independent" samples. Qualitatively, we may say that if the Markov Chain forgets the past
samples fast enough, then we may sooner treat them as independent from each other. Gelman (2014) suggests that in order to assess this quantitatively, multiple MH-chains could be run in parallel and statistical quantities within and between each chain are analysed. For a detailed discussion thereof, we refer the reader to Appendix C.

Once a satisfying number of samples has been drawn from the posterior pdf, a marginalisation of the nuisance parameters can be done by looking at the parameters of interest only. These samples may then be treated like counts in a histogram, i.e.
distributional estimates, or simply the interesting statistical moments, such as mean and variance, can be obtained.





### 3.3 Construction of the bedrock-ice interface

The main analysis programme allows us to export all parameters either as a full chain dataset, where every single draw is recorded, or as a statistical summary (i.e. mean and variance). Both are then converted to point data, i.e. (x, y, z) – data. For the subsequent construction of the interface between rock and ice (see section 5.2 and Fig. 1 for the presentation of the test 525 experiment) we only need the full-chain point-data. In the present study we restrict ourselves to a probabilistic description until the bedrock positions within a cone. It would also be possible to treat the bedrock construction within a Bayesian framework, however this would go beyond the scope of this study and is therefore left for a future adaption of the code. Nonetheless, in order to construct a surface, we rely on deterministic methods, which are explained in detail in what follows.

### 3.3.1 Interpolation to a grid

The "modelviewer.py" routine is able to read datasets from different detectors (which are saved as JSON-files) and computes for each cone the statistic, which the user is interested in (see "sigma" entry in program). Thus, it is possible to use the mean or, for example, the $+1\,\sigma$ position of each cone. From here onwards this point cloud is named $H$ and contains one interface position (x, y & z coordinates) per cone. These are shown as triangles (▲) in Fig. 5.

As a second step, the programme interpolates this point cloud in a bilinear way to a rectangular grid with a user specified cell 535 size, $\Delta_{cs}$. This grid can be described by a matrix $P \in \mathbb{R}^{r \times c}$, where $r$ and $c$ are the number of rows and columns (i.e. the number of y- and x-cells, needed to cover the whole grid). The procedure is similar to the bilinear interpolation of Lagrangian markers (that carry a physical property) to a (fixed) Eulerian grid in geodynamical modelling (see Gerya, 2010, p. 116), with the difference that our physical property is the height of the ice-bedrock interface (Fig. 1).

We could also have fitted a surface through the resulting point cloud. However, by formulating this surface as a matrix we 540 gain access to the whole machinery of linear algebra. Moreover, $P$ can directly be interpreted as a rasterised DEM, which can be easily loaded and visualised in any GIS software. Thus, from a modular design perspective we think that the matrix formulation has more advantages than drawbacks. The bilinear interpolation is shown in more detail in Fig. 5.



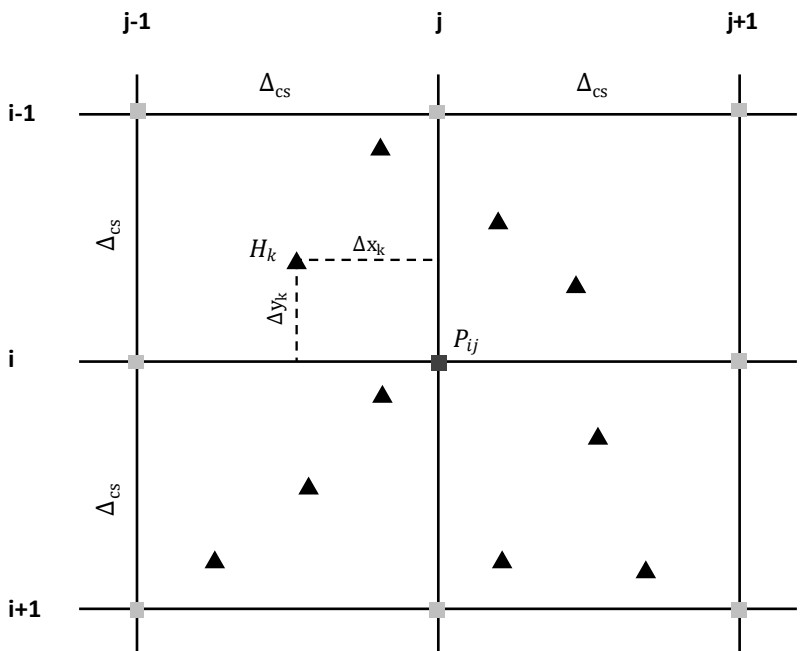

**Figure 5: Two-dimensional stencil, used to summarise the bilinear interpolation of interface positions within cones ($H_k$, ▲) to a fixed grid ($P_{ij}$, ■) with a user-defined cell size $\Delta_{cs}$. Every interface position within a $\pm\Delta_{cs}$ interval contributes to the grid height $P_{ij}$.**

In order to calculate the height at a grid point, $P_{ij}$, one has to form a weighted sum over the entire cone interface positions within a $\pm\Delta_{cs}$ interval, i.e.

$$P_{ij} = \frac{\sum_k w_k H_k}{\sum_l w_l},$$ (38)

where the weights, $w_k$, are given by

$$w_k = \left(1 - \frac{\Delta x_k}{\Delta_{cs}}\right) * \left(1 - \frac{\Delta y_k}{\Delta_{cs}}\right).$$ (39)

### 3.3.2 Damping & Smoothing

The concept of damping usually revolves around the idea to force parameters to a certain value (e.g. in deterministic inversion by introducing a penalty term in the misfit function for deviations from that value). From a Bayesian viewpoint this would be accomplished by setting the prior mean to a specific value. In our code we implemented this idea by allowing the user to read a DEM and a "damping weight" to the code (see "fixed length group" in code). The programme effectively computes a weighted average between the bedrock positions within the cones and a user defined DEM. The higher the chosen damping weight, the more the resulting interface will match the DEM, when pixels overlap.





The matrix formulation also enables us to use a further data processing technique without much tinkering. As geophysical
data are often quite noisy, a standard procedure in nearly every geophysical inversion is a smoothing constraint. This
effectively introduces a correlation between the parameters and forces them to be similar to each other. From a Bayesian
perspective, we could have achieved this correlation by defining a prior covariance matrix of the thickness parameters, such
that neighbouring cones should have similar thicknesses (which makes sense as we do expect the bedrock-ice interface to be
relatively continuous; Fig. 1). As we work with independent cones in this study, we leave the exploration of this aspect open
for a future study. Nevertheless, we offer the possibility in our code to use a smoothing on the final interpolated grid. This is
achieved by a convolution of a smoothing kernel, $K$ (see Appendix D for details), with the surface matrix $P$, which results in
a smoothed surface matrix

$$P^S = K * P. \tag{40}$$

Please note that the $*$ operator in Eq. (40) denotes a convolution. In index notation the advantage of the linear algebra
formalism becomes clear, as $P^S$ can be expressed by

$$P_{ij}^S = \sum_{k=-s}^{s} \sum_{l=-s}^{s} K_{k+s+1,l+s+1} P_{i+k,j+l} \,. \tag{41}$$

The user is free to choose the number of neighbouring pixels, $s$, across which the programme performs a smoothing. As a
related matrix we use an approximation to a Gaussian kernel, which corresponds to a Gaussian blur in image processing.
Whereas "smoothing" is a general term used in the geophysical community where such a process is used with an aim to
force a correlation of parameters, in our case where the parameters describe a surface (Fig. 1), the convolution effectively
smooths the surface, i.e. removes small-scale variations.

Finally, we added a checkbox to our code to allow it to change the order of the damping and smoothing operations.
Sometimes when a strong damping is necessary, this may result in rather unsmooth features at DEM boundaries, such that it
makes sense to perform a smoothing only afterwards.

## 4 Main modules of SMAUG

Our toolbox, SMAUG, contains several subprograms, which are executed separately. This allows the user to inspect
intermediate results without any difficulty. We also tried to keep the intermediate results as portable as possible, by using
JSON-files, as often as possible. Here we explain, in logical order, the rational of the submodules (a detailed user manual is
separately available):


***MATERIALIZER.py***



This subroutine allows the user to create their own material that will be used in the subsequent model builder. The user may choose a density (either from data or directly insert mean and standard deviations) and a composition (also either from data or from the list of Groom et al., 2001).


### DATA_BINNING.py

As the name suggests, this subroutine is used to spatially bin the recorded track data. The bin data (i.e. the output hereof) is then fed to the model builder.

### MODEL_BUILDER.py

The model builder takes the bin data and the materials as inputs and allows the user, with help of DEMs, to allocate data and materials to certain cones. This is basically the spatial setup of the model. The resulting model file is then provided to the inversion code.

### INVERSION.py

This is the main module in SMAUG, providing the functionality to perform a MCMC algorithm on the probabilistic model created with MODEL_BUILDER.py. It also includes several analysis tools to assess MCMC performance.

### MODEL_VIEWER.py

The model viewer allows us to visualise the interface results, obtained and exported by INVERSION.py. It also has the functionality to dampen and smooth the resulting surfaces.





## 5 Model Verification

In this section we present examples of how the model can be employed, what it calculates and how the output is structured. We proceed by verifying, in a first step, that the physical models employed in this work yield results which are numerically consistent with the results of calculations from other studies. We will compare our results with reported values from literature in Ch. 5.1. We do this because we do not change the parameters of the flux model (except the height scaling, which has been verified by Nishiyama et al. (2017), and since the energy loss calculations is based on equations that stem from

different frameworks.

In a second step, we will benchmark the results obtained by this code from real data against previously published results. For this purpose, we will re-analyse the raw data from Nishiyama et al. (2017). This is a study (see also Fig. 1) that was conducted in the Central Swiss Alps in a railway tunnel that featured a glacier (part of the Great Aletsch glacier) above. Our goal there was to estimate the thickness of the overlying glacier and thus the subsurface structure of the ice-bedrock

interface. The respective calculation and discussion thereof are presented in Ch. 5.2.

### 5.1 Verification of energy loss calculations

The energy loss model that we use in our code generally reproduces the literature values well (below 1%) across the different energy loss processes and relevant energies. In Fig. 6 we present the energy loss calculations for each energy loss process (i.e. ionisation, bremsstrahlung, pair-production and photonuclear interactions) across energies from 10 MeV to 100 TeV for

silicon.

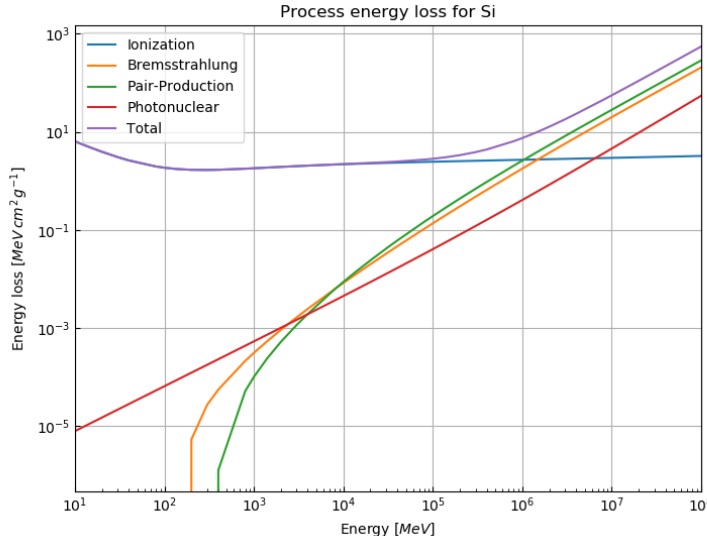

**Figure 6: Log-log plot of the stopping power of the different energy loss processes for silicon. At ~ 10 GeV the radiative processes (i.e. bremsstrahlung, pair-production and photonuclear interactions) reach around 1 % of the total stopping power. At a few hundred GeV (at the so called "critical energy") the radiative processes start to become dominant over the ionisation losses.**





The overall characteristics between the different elements are the same with minor differences regarding the position of the critical energy and the 1 % - radiative point. In Fig. 7 we show the relative error of our calculations to the tabulated values from Groom et al. (2001) for the whole energy range.

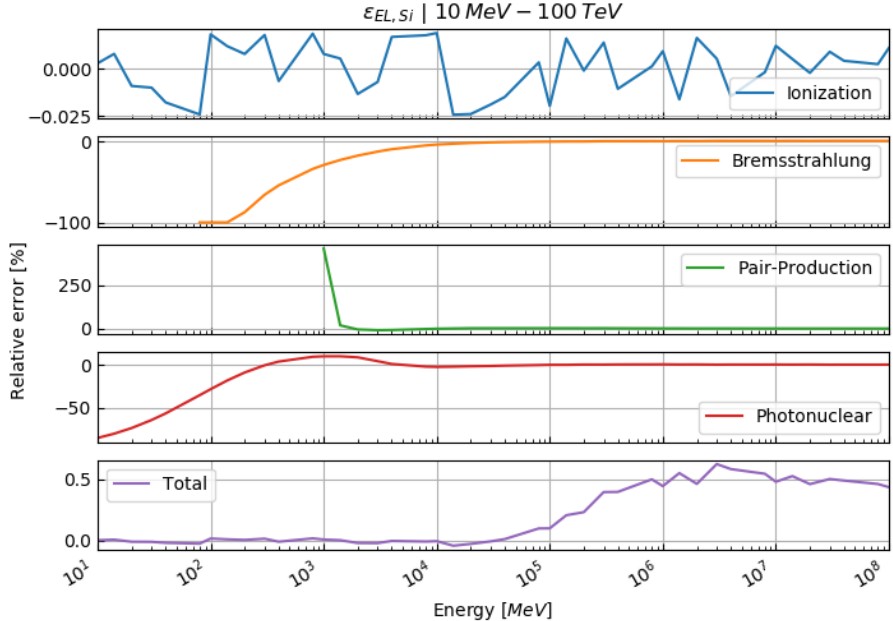

**Figure 7: Relative error of our energy loss calculations compared to the tabulated values from Groom et al. (2001) for silicon.**
**Ionisation losses agree very well with the literature values (within 0.025 %). At low energies the relative errors of the radiative processes are large and converge to a value close to 0 towards higher energies, resulting in a relative error on the total energy loss of around 0.5 % compared to literature.**

We note that the energy losses by ionisation are reproduced very well over the entire energy range. We also note that the relative error on the radiative energy losses is rather large below 10 GeV. This does, however, not introduce a major bias,
because below this energy, radiative energy losses are negligible compared to ionisation losses, as can be seen in Fig. 6. Furthermore, the related errors are in an acceptable range at the energy level at which radiative losses begin to become noticeable (i.e. around 100 GeV). This can be seen in Fig. 7, in the sense that the total relative error remains well bounded within 0.5 %. In the ionisation domain (i.e. below 100 GeV) the total relative error is dominated by the ionisation relative error, whereas above this energy level the relative errors on radiative losses start to prevail. A close-up of this energy range
is given in Fig. 8.

There are different sources and circumstances that contribute to the error in the different energy losses processes. The scatter of the relative ionisation-loss error around 0 with a rather small deviation can be viewed as simple rounding errors. The errors on the radiative processes, however, seem to be of a more systematic nature. We explain this behaviour through a





different numerical integration scheme in Eq. (7), which tends to systematically under-/overestimate the true value,
especially when the integrand comprises exponential functions. Whereas we used a Double Exponential Integration scheme
(see Takahasi & Mori, 1974), the integration scheme from Groom et al. (2001) is not discernible. However, as the relative
errors on the processes of energy loss remain well within the theoretical uncertainties, (see Ch. 3.1.3), we consider, that our
calculation accurately reproduces the literature values for elements.

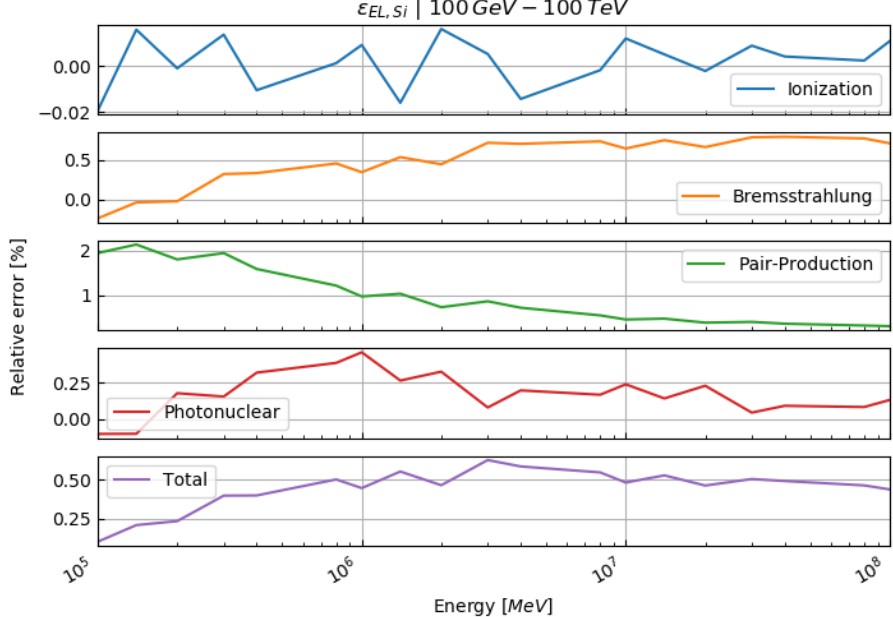

**Figure 8: Relative error of our energy loss calculations for silicon compared to the tabulated values from Groom et al. (2001) at higher energies (100 GeV – 100 TeV). The relative errors remain bounded within their theoretical uncertainties (see Ch. 3.1.3).**

The above calculations were performed for pure silicon. The respective figures for other four important elements in the
Earth's crust (Al, Fe, Ca & O) can be found in Appendix E. Those elements are, however, not representative for any real
material encountered in geological applications. For this reason, we compiled the same computations for four selected,
geologically important compounds (SiO2, CaCO3, Standard Rock, ice) that are also shown in Appendix E. We summarise,
that with the exception of Standard Rock, all calculations yield results that are similar to the silicon calculation above. The
discrepancy for Standard Rock stems from its inconsistent definition, with respect to the different parameters. In particular,
the "Standard Rock" according to Lohmann et al. (1985) has an atomic number Z of 11 (i.e. sodium) and an atomic weight A
of 22, which yield the characteristic parameter values of $\langle Z/A \rangle = 0.5$ and $\langle Z^2/A \rangle = 5.5$ respectively. Note that Groom et al.
(2001) list sodium as the only constituent of a standard rock. However, this material cannot be modelled by any mixture of
pure elements, as common sodium consists of one neutron more and thus has a higher atomic weight (i.e. $A_{Na} = 23$).





Consequently, the use of standard sodium would lead to different characteristic parameter values, i.e. $\langle Z/A \rangle = 0.478$ and $\langle Z^2/A \rangle = 5.263$, thus leading to an inconsistency. This is often conveyed by the phrase that standard rock "is not-quite-sodium" (Groom et al. 2001, p.203). In order to circumvent this problem, we advocate the exchange of $^{23}_{11}Na$ with its $^{22}_{11}Na$

isotope, that would lead to the characteristic parameter values $\langle Z/A \rangle = 0.500$ and $\langle Z^2/A \rangle = 5.501$, which are much closer to the actual definition of standard rock. For this reason, we extended the element/compound-list, (which is available from http://pdg.lbl.gov/2019/AtomicNuclearProperties/expert.html) by the $^{22}_{11}Na$ isotope, assuming that all parameters are equal to the ones from $^{23}_{11}Na$. Additionally, we redefined the standard rock (i.e. material number 281 in the list) to consist only of $^{22}_{11}Na$. With this change, standard rock does not need any more special treatment and can be calculated in a way that is

consistent to all other compounds. Furthermore, the relative error between the tabulated values and our modified calculation falls in line with the calculations for the other compounds and elements (Figs. 9 and 10).

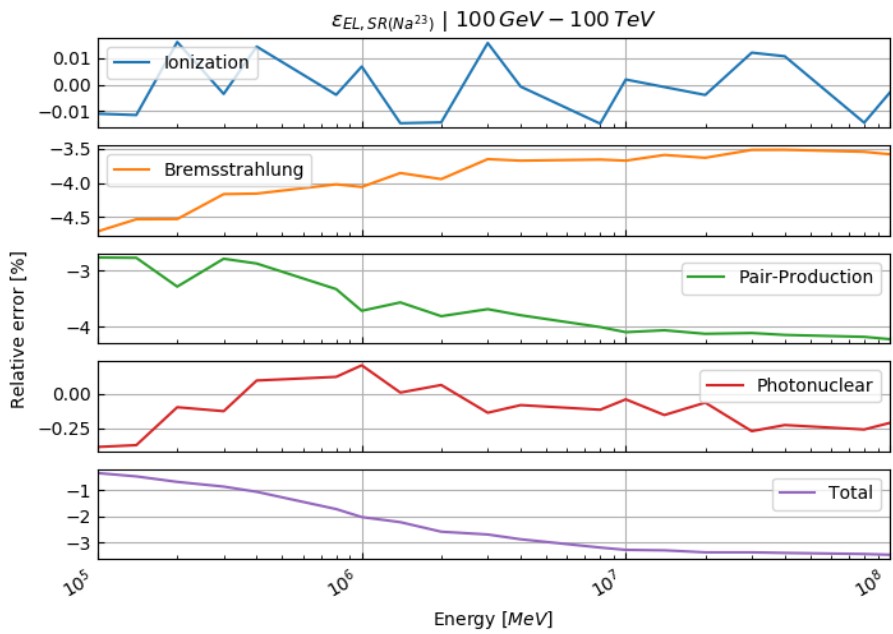

**Figure 9: Relative error of our energy loss calculations for a standard rock consisting of $^{23}_{11}$Na, compared to the tabulated values from Groom et al. (2001) at higher energies (100 GeV – 100 TeV).**






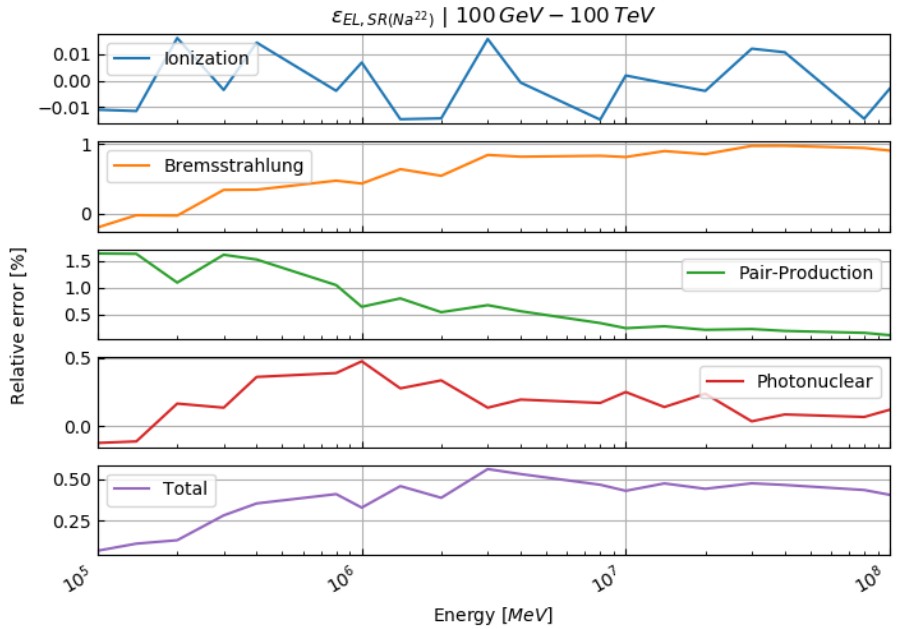

**Figure 10: Relative error of our energy loss calculations for a standard rock consisting of $^{22}_{11}$Na, compared to the tabulated values from Groom et al. (2001) at higher energies (100 GeV – 100 TeV).**

### 5.2 Verification of the bedrock-ice interface reconstruction

In this part we test the presented reconstruction algorithm on previously published data. For this purpose, we compare our calculations to the ones already published in the study by Nishiyama et al., (2017), where the goal was to measure the interface between the glacier and the rock, in order to determine the spatial distribution of the rock surface (also below the glacier). We could access the Railway Tunnel to install the muon detectors beneath the ice. A situation sketch is shown in Figs. 1 and 11.

The results shown below (Figs. 12 – 14) represent the bedrock-ice interface interpolated to an 8-metre grid, which was first damped (weight 8) and then smoothed (2 grid pixel). We assess the goodness of fit according to the three cross-sections (East, Central, West), that are shown in Fig. 10. The crosscuts are nearly perpendicular to the train tunnel and roughly 40 $m$ apart from each other. Figures 12 to 14 show the three cross-sections in detail. In every plot, we also indicate the solution from Nishiyama et al. (2017). Please note that we added a systematic error of 2 $m$ to the uncertainty planes, as the DEM we

are working with has itself an uncertainty of ± 2 $m$. The dash-dotted lines mark thus the most conservative error estimate. Moreover, we highlighted the parts of the cross-section that had either been damped to the bedrock DEM or that have been solely resolved by the measurement (see "damping marker" in Fig. 11).

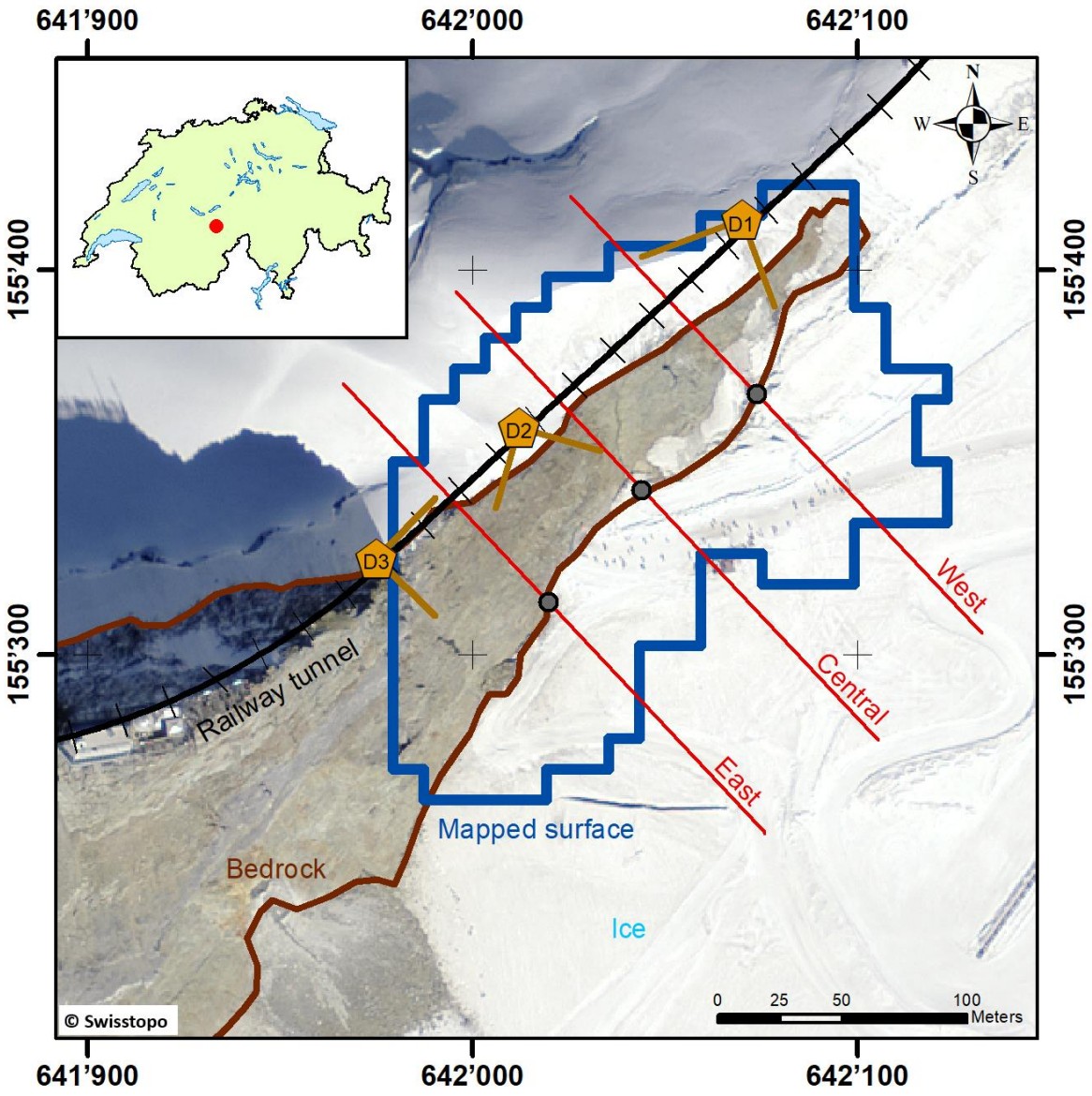

**Figure 11: Overview map of the study area at Jungfraujoch. The brown line separates the visible bedrock in the DEM from the glacier part ("Ice"). The three profiles (East, Central & West) are depicted with a red line, on which the damping marker is shown by a grey point. The extent of the reconstructed bedrock-ice interface is shown by the blue area. Additionally, the three detector positions (D1, D2 & D3) are shown by orange pentagons, including their viewfield. Basemap: Orthophotomosaic Swissimage, © Federal Office of Topography swisstopo.**

Figure 12 shows the western profile, where our bedrock-ice interface and the one from the previous study agree well and

both lie within the given error margins. The lack of fit in areas where the steepness changes rapidly (i.e. around $40\,m$ and





80 $m$) can be explained as a smoothing artefact. Towards the end of the profile, the decreasing data coverage becomes evident as the uncertainties rise. This effect can also be seen in the jagged behaviour of the interface curves around 100 $m$ to 120 $m$, hinting at the effect where the interpolation has occurred with few data.


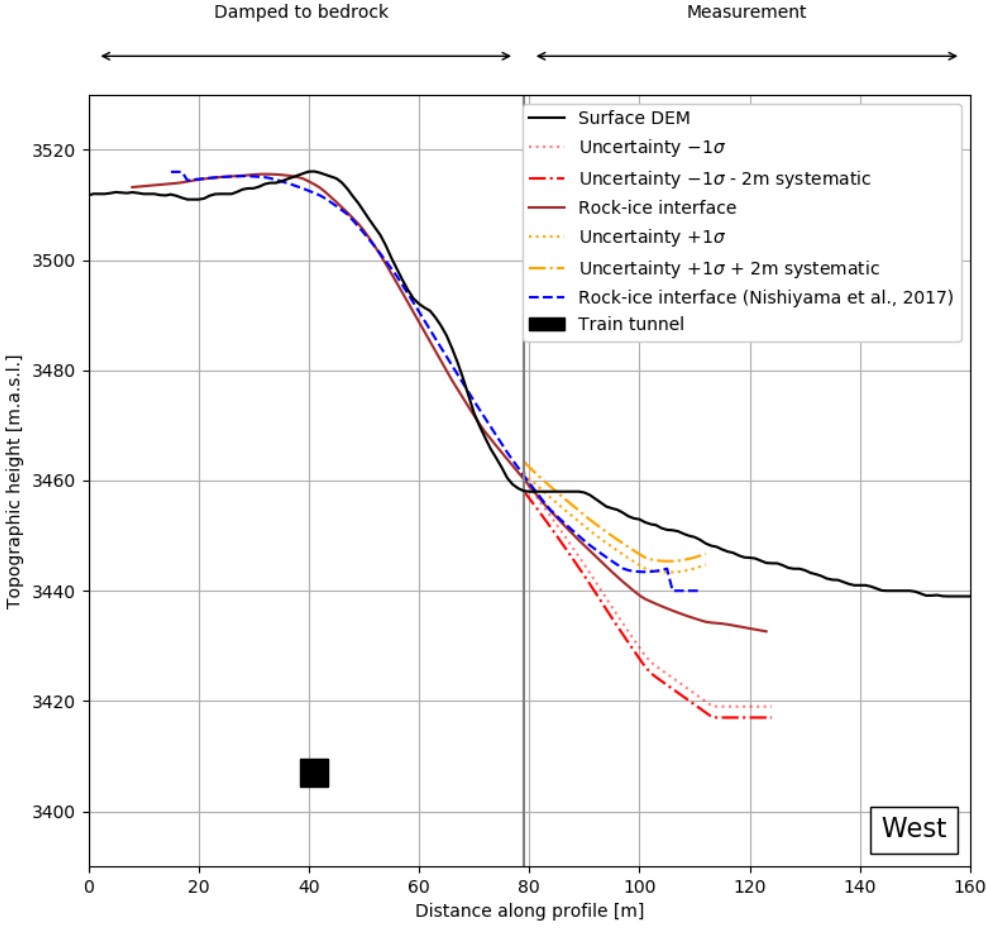

**Figure 12: Western cross-section. The brown and dashed blue line indicate the ice-bedrock interface solutions of this study and the one from Nishiyama et al. (2017), respectively. 1 $\sigma$-error margins are shown in yellow (upper) and red (lower). The dotted margins encompass only the statistical variation of the interface position, whereas the dash-dotted include a $\pm 2$ $m$ systematic error which stems from the inherent DEM-uncertainty. For completeness we also show the position of the railway tunnel as a black square.**


Figure 13 presents the central profile. Similar to the western profile (Fig. 12) the fits match quite well and are within the error margins. It may be possible that the point where the actual bedrock begins might be further down (i.e. ~80 $m$ instead




of 65 $m$). Here we used the same DEM and aerial photograph as in the previous study. This means that newer versions might

be available that show more bedrock (due to the glacial retreat as a response to global warming).

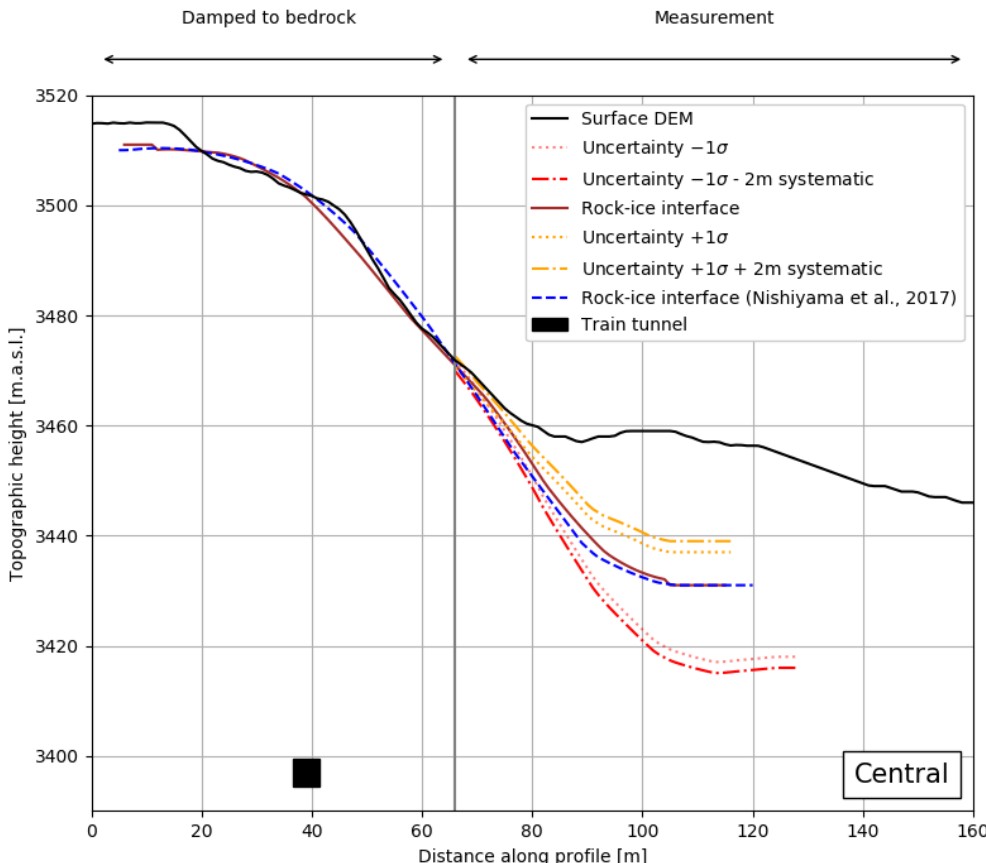

**Figure 13: Central cross-section. The brown and dashed blue line indicate the ice-bedrock interface solutions of this study and the**
**one from Nishiyama et al. (2017), respectively. 1 $\sigma$-error margins are shown in yellow (upper) and red (lower). The dotted**
**margins encompass only the statistical variation of the interface position, whereas the dash-dotted include a $\pm 2\,m$ systematic**
**error, which stems from the inherent DEM-uncertainty. For completeness we also show the position of the railway tunnel as a**
**black square.**

The eastern profile is shown in Fig. 14. One sees that the results from this study are internally consistent. The surface from

the previous study plunges down earlier with respect to the surfaces calculated here. This may in fact be a damping effect, as

the bedrock-ice interface from Nishiyama et al. (2017) has not been constrained to the bedrock (via damping) and thus

plunges down before the damping mark at $\sim 72\,m$. Still, the two surfaces agree within 5 $m$, which we consider as acceptable.





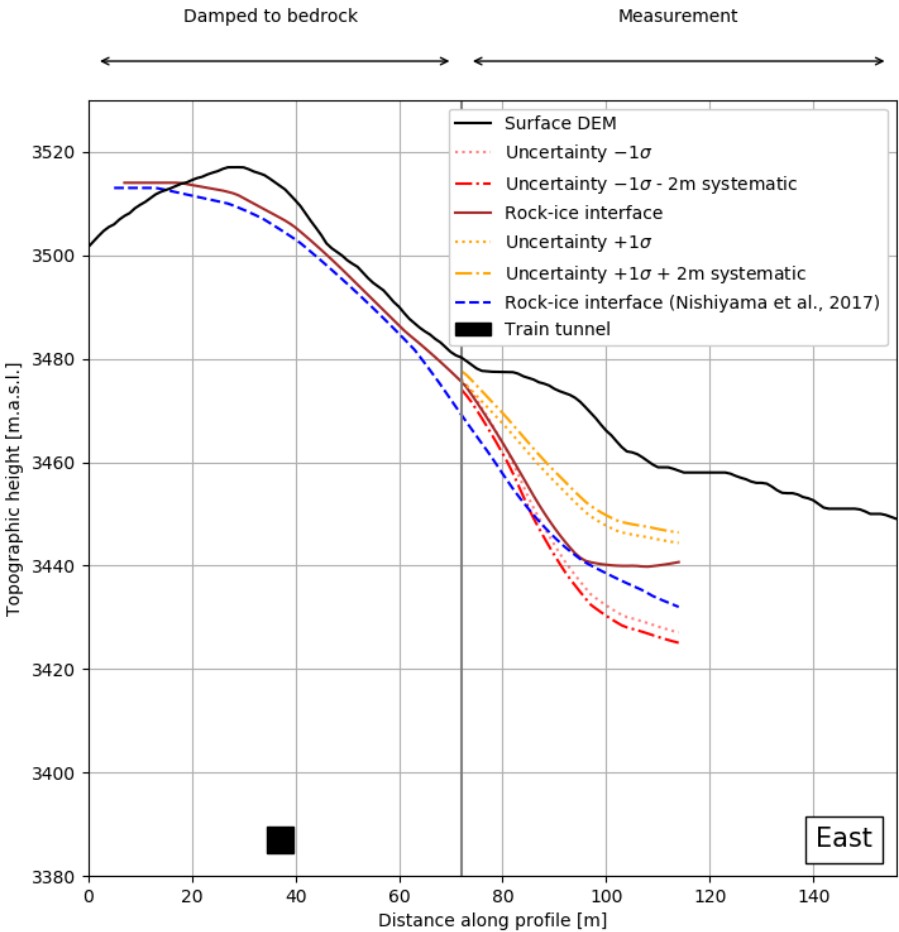

**Figure 14: Eastern cross-section. The brown and dashed blue line indicate the ice-bedrock interface solutions of this study and the one from Nishiyama et al. (2017), respectively. 1 $\sigma$-error margins are shown in yellow (upper) and red (lower). The dotted margins encompass only the statistical variation of the interface position, whereas the dash-dotted include a $\pm 2\,m$ systematic error which stems from the inherent DEM-uncertainty. For completeness we also show the position of the railway tunnel as a black square.**

All together the performance of the whole workflow, which is shown in this study, produces results, which are similar to the ones published in the previous study (Nishiyama et al., 2017). We use the results of this comparison to validate the base of our code.







## 6 Conclusion

In this study we have presented a model that allows us to integrate geological information into a muon tomography framework. The inherent problem of parameter estimation has been formulated in a probabilistic way and solved accordingly. The propagation of uncertainties thus occurs automatically within this formalism. We also considered approaches including DAGs or the simplex subspace of compositions which could be helpful to the muon tomography community while tackling their own research. We condensed these approaches in a modular toolbox. This assortment of python programmes allows the user to address the subproblems during the data analysis of a muon tomography experiment. The programmes are modular in the sense that the user can always access the intermediate results, as the files are mostly in a portable format (JSON). Thus, it is perfectly possible to only use one submodule of the toolbox while working with an own codebase. As every "tool" is embedded in a GUI, the programme is made accessible without the need to first read and consider several thousand code lines. Furthermore, we have shown that the results we obtain with our code are largely in good agreement with an earlier, already published experiment. The small deviations may be attributed to data analysis subtleties.

We would like to stress that this work is merely a foundation upon which many extensions can be built when it is used in other applications as well. Future content might, for example, include a realistic treatment of multiple scattering and the inclusion of compositional uncertainties in the inversion, for which we laid out the basis in this study.





**Appendix A – Muon flux model**

As many empirical muon flux models, the one that we employed consists of an energy spectrum for vertically incident muons at sea level at its core. An accepted instance is the energy spectrum of Bugaev et al. (1998), that takes the form

$\quad \Phi_B(p) = A_B \, p^{-(\alpha_3 \log_{10}^3(p) + \alpha_2 \log_{10}^2(p) + \alpha_1 \log_{10}(p) + \alpha_0)},$ (A1)

where $p$ denotes the momentum of the incident muon in $GeV * c^{-1}$. The values of the $\alpha_i$ and $A_B$ are, for example, listed in Lesparre et al. (2010). This model is an extended version of Renya (2006), to account for different incident angles,

$\quad \Phi_R(p, \theta) = \cos^3(\theta) \, \Phi_B(p \cos(\theta)),$ (A2)

where $\theta$ is the zenith angle of the incident muon. It is important to note that the parameter values in Eq. (A1) are changed to 770 $\quad \alpha_0 = 0.2455, \alpha_1 = 1.288, \alpha_2 = -0.2555, \alpha_3 = 0.0209$ and $A_B = 0.00253$. In order to include height above sea level as an additional parameter, Hebbeker and Timmermans (2002) proposed to model the altitude dependency as an exponential decay, which modifies Eq. (A2) into

$\quad \Phi(p, \theta, h) = \Phi_R(p, \theta) * \exp\left(-\dfrac{h}{h_0}\right).$ (A3)

The scaling height, $h_0$, is usually to be taken as $h_0 = 4900m + 750 \, m \, c \, GeV^{-1} * p$, where $p$, is the momentum of the 775 incident muon in $GeV * c^{-1}$. However, as this formula is only valid up to an altitude of 1000 m above sea level, Nishiyama et al. (2017) adapted it to $h_0 = 3400 \, m + 1100 \, m * c * GeV^{-1} * p * \cos(\theta)$. This was done in order to fit the energy spectrum up to 4000 m above sea level. This formula is now valid for momenta above $3 \, GeV * c^{-1}$, zenith angles between $0°$ and $70°$ and an altitude below $4000 \, m$ above sea level.

**Appendix B – Rock model**

**B1 – Density model**

The density distribution of a lithology can be determined through various methods. In our work, we constructed a density model by analysing various rock samples from our study area in the laboratory. Two experimental setups were employed to gain insight into the grain, skeletal as well as the bulk density of the rocks. Grain and skeletal density were measured by means of the AccuPyc 1340 He-pycnometer, which is a standardised method that yields information on the volume. Bulk 785 density values were then determined based on Archimedes' principle, where paraffin coated samples were suspended into water (ASTM C914-09, 2015; Blake and Hartge, 1986).

Every sample $j = 1, \dots, N$ (usually the size of a normal hand sample) has been split up into smaller subsamples $i = 1, \dots, S_j$, that were measured. The bulk density of the i-th subsample can be calculated by





$$\rho_{bulk,ij} = \frac{\rho_{H_2O} * m_{s,ij}}{(m_{s,ij} + m_{p,ij} + m_{t,ij} - m_{sus,ij}) - \left(\frac{m_{p,ij} * \rho_{H_2O}}{\rho_p}\right) - \left(\frac{m_{t,ij} * \rho_{H_2O}}{\rho_T}\right)}, \qquad \text{(B1)}$$

where $\rho_{H_2O}, \rho_p, \rho_T$ denote the density of water, paraffin and the thread that was used to dip the sample into the liquid, respectively. $m_{s,ij}, m_{p,ij}, m_{t,ij}, m_{sus,ij}$ describe the mass of the sample, the paraffin coating, the thread and the apparent mass of all three components suspended in water. $m_{p,ij}, m_{t,ij}$ can then be simply obtained through

$$m_{p,ij} = m_{s,t,p,ij} - m_{s,t,ij}, \qquad \text{(B2)}$$

as $m_{s,t,p,ij}$ denote the mass of the sample including thread and paraffin coating on one hand and $m_{s,t,ij}$ only the mass of the
sample and the thread on the other hand. Further, the mass of the thread is given by

$$m_{t,ij} = m_{s,t,ij} - m_{s,ij}. \qquad \text{(B3)}$$

The maximal precision of the reading is estimated at $\pm 5 * 10^{-5} g$, and the commonly ignored effects regarding buoyancy in air has been estimated to introduce an error on the order of $\pm 2 * 10^{-4} g$. This error has been attributed to all direct mass measurements. Moreover, because small pieces of material may detach from the sample upon attaching the thread to the
sample and during the paraffin coating, we set an error of $\pm 2 * 10^{-2} g$ to all measurement results. The variables in Eq. (B1) are strictly positive values. Following Tarantola (2005) we model these "Jeffreys parameters" by lognormal distributions, as they inherently satisfy the positivity constraint. Because Eq. (B1) does not simply allow a standard uncertainty propagation, the script "subsample_analysis.py" performs a Monte Carlo simulation for each subsample and attributes a final lognormal probability density function to the resulting histogram. Figure B1 illustrates such an example, where the calculation has been
performed for subsample JT-20-1.

We have found 10'000 draws per subsample to be sufficient to retrieve a solid final distribution. However, this parameter can easily be changed in the script, depending on the user's preference of precision/speed. From this point onwards we may work with a Gaussian distribution as Fig. B1 assures us that a normal pdf describes the results of the Monte Carlo simulation rather well.

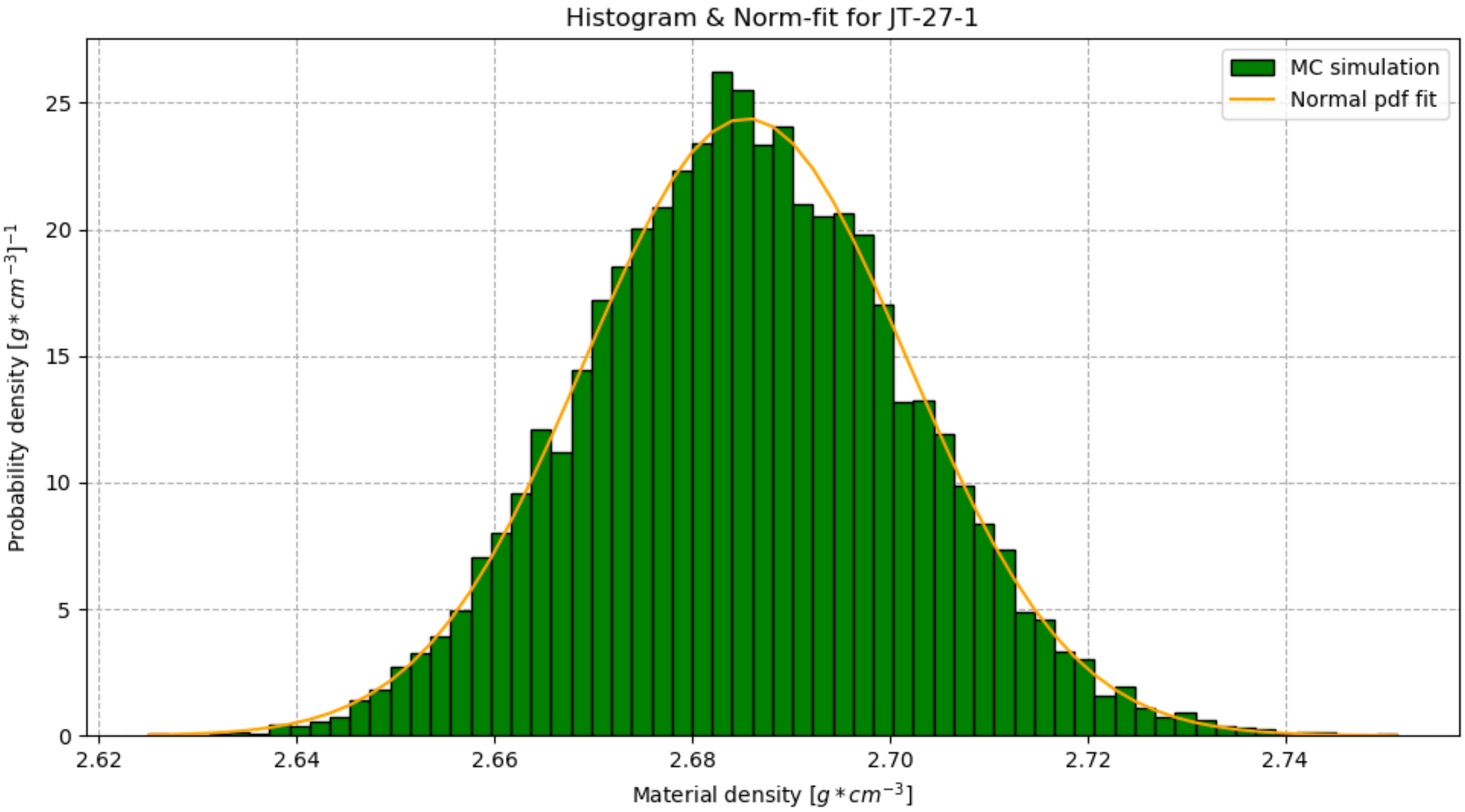

**Figure B1: Example output of "subsample_analysis.py" for a bulk density measurement of subsample JT-27-1 (see supplementary information for data). Green bars represent the histogram of 10'000 Monte Carlo simulation draws. The orange curve indicates the fitted normal probability density function.**

The determination of the grain and skeletal densities is simpler than the bulk density measurements because the corresponding method consists of a mass and a volume measurement, respectively. The density formula reads then simply

$$\rho_{skeletal/grain,ij} = \frac{m_{ij}}{V_{ij}}. \tag{B4}$$

The question remains as to how it is possible to construct a pdf that represents the knowledge about the whole lithology. There are two possible methods that can be readily employed at this point. The first, following largely Tarantola (2005), performs a so-called disjunction of the pdfs that corresponds to an averaging of all subsample pdfs. As Vermeesch (2012) points out, even though this might seem a "sensible strategy at first glance", there might be some problems with this method. The main problem lies in the small error on the subsamples, such that the variation between different subsamples may be larger than their attributed errors. This would not be a problem if enough subsamples could be measured, such that the resulting lithology pdf might be sampled correctly. On the other hand when one is faced with a situation where data is rather scarce, then the approach of Tarantola (2005) would result in a rather spikey pdf that would be hard to handle. For this reason we adopted the methodology of Vermeesch (2012) where the lithology pdf is estimated by a kernel density

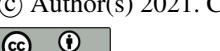



estimation. The main difference lies in the fixed "bandwidth" of the subsample distributions. We refer to Vermeesch (2012) for more details and an in-depth discussion on this problem.

The kernel density estimation has the advantage, that only the mean values of the subsamples have to be processed as the bandwidth is determined from the spread of the subsample means. Following the methodology of Vermeesch (2012) we end up with a pdf like the one visualised in Fig. B2. We could at this point use the kernel density estimated pdf for further calculations. However, for simplicity we approximate the kde with a normal distribution and intend to add support for the kde in a later code version.

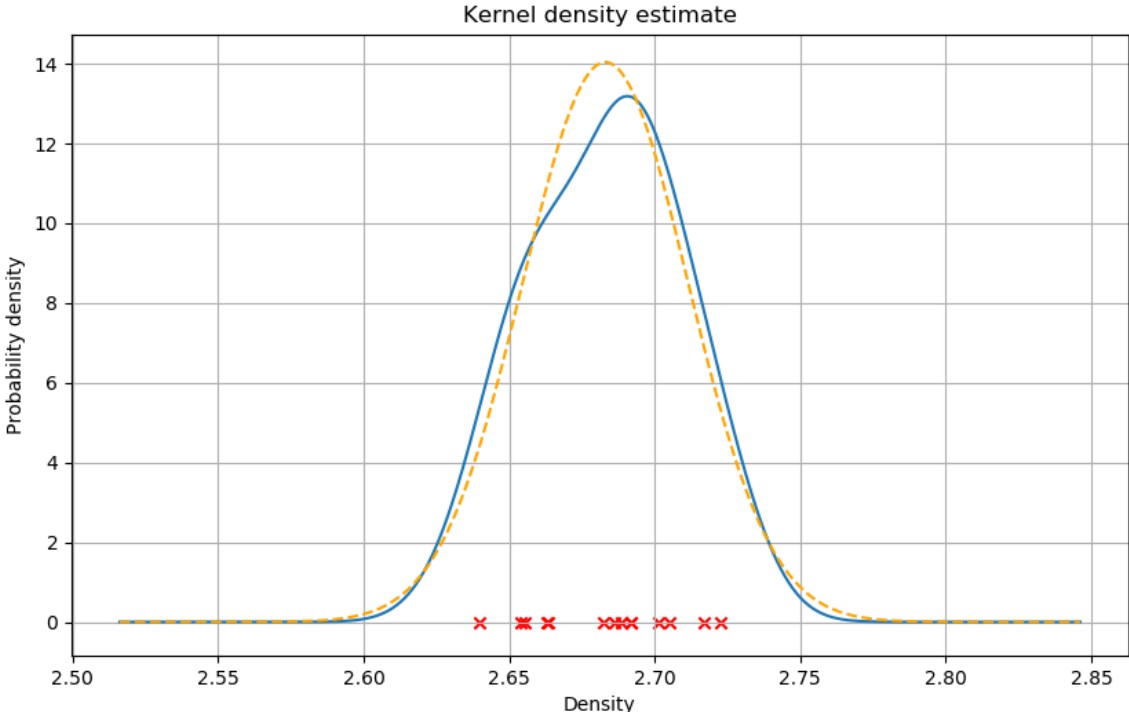

**Figure B2: Example output of "materializer.py". Here a set of subsample mean values (red crosses) are processed in a kernel density estimate (solid blue line). Finally a normal distribution is fitted to the kernel density estimate (dashed yellow line).**




## B2 – Composition model

We have seen in Eq. (9) that the material density parameter enters the energy loss calculations rather directly. Contrariwise, the compositional model affects the energy loss equations much more subtly through the average $\{Z/A\}_{rock}$ and $\{Z^2/A\}_{rock}$
values and mean excitation energies that need to be calculated for the entire lithology. Likewise, information on the weight percentages of the main elements within the rock is required for the quantification of the radiation loss term.

Although a modal mineral analysis (e.g. the quantitative determination of mineral volumes) is preferable and can be treated according to Lechmann et al. (2018), its execution is a rather time-consuming effort. This is the reason why compositional data in muon tomography experiments predominantly consist of XRF-data, which show the abundance of major oxides
within the rock. We describe here a method to incorporate such type of information in a probabilistic way thereby following Aitchison (1986). Compositional data are usually available in the form of Table B1, which presents an excerpt of four samples for illustration purposes. We refer to the excel sheet in the supplementary material of the present work for the full data.

**Table B1: Excerpt of XRF data for four samples. Data in column denote weight percentages of major oxides within the rock samples.**

| Sample | JT01 | JT02 | JT19 | JT20 |
|---|---|---|---|---|
| Oxides | | | | |
| SiO2 | 0.6131 | 0.5981 | 0.6997 | 0.6139 |
| TiO2 | 0.0123 | 0.0067 | 0.0076 | 0.0094 |
| Al2O3 | 0.1567 | 0.1873 | 0.1481 | 0.1921 |
| Fe2O3 | 0.087 | 0.0791 | 0.0496 | 0.0686 |
| MnO | 0.001 | 0.0012 | 0.0009 | 0.0009 |
| MgO | 0.0359 | 0.0285 | 0.0206 | 0.0288 |
| CaO | 0.0202 | 0.0071 | 0.0201 | 0.0137 |
| Na2O | 0.0228 | 0.0248 | 0.0404 | 0.0323 |
| K2O | 0.0343 | 0.0465 | 0.0287 | 0.0469 |
| P2O5 | 0.0041 | 0.0029 | 0.0021 | 0.0027 |
| Sum | 0.9874 | 0.9822 | 1.0178 | 1.0093 |

There are several challenges to this kind of data. First, the parameters (i.e. the oxide percentages) can take a value between 0 and 1. This means that normal as well as lognormal distributions are not suitable to describe these parameters. Second, the
requirement that the sum of all parameters has to ideally equal 1 poses a constraint on this parameter space, which effectively reduces the number of independent parameters by one. Third, due to measurement uncertainties, this sum is never exactly one.



Spaces, which have this unit sum condition can be viewed as a simplex, e.g. if we had three compositional parameters, the simplex would be a 2-dimensional surface (i.e. a subspace) in this 3-dimensional parameter space. The last issue, of not
summing up exactly to 1, can be remedied by projecting each sample dataset back to the simplex (Aitchison, 1986, p. 257-261). This works only if the measurement imprecisions are not too large, which works well for the examples in Table B1. With respect to the energy loss calculation, it is preferable to decompose the oxides into elements, which can be done by following formula

$$wt_{ele,i} = \sum_{j \in \{oxides\}} wt_j * \frac{n_{ij} m_i}{m_j}, \tag{B5}$$

where $m_i$ and $m_j$ denote the molar mass mass of the i-th element and the j-th oxide, $wt_j$ is the j-th datum in the column and $n_{ij}$ is the number of atoms of the i-th element within the j-th oxide. The two transformations are visualised in Table B2.





**Table B2: Element weight percent data. Transformed from oxide weight percent data with use of Eq. (B5). All data has additionally been scaled to satisfy the unit sum constraint.**

| Sample | JT01 | JT02 | JT19 | JT20 |
|---|---|---|---|---|
| Elements | | | | |
| Si | 0.2902 | 0.2846 | 0.3213 | 0.2843 |
| Ti | 0.0075 | 0.0041 | 0.0045 | 0.0056 |
| Al | 0.0840 | 0.1009 | 0.0770 | 0.1007 |
| Fe | 0.0616 | 0.0563 | 0.0341 | 0.0475 |
| Mn | 0.0008 | 0.0009 | 0.0007 | 0.0007 |
| Mg | 0.0219 | 0.0175 | 0.0122 | 0.0172 |
| Ca | 0.0146 | 0.0052 | 0.0141 | 0.0097 |
| Na | 0.0171 | 0.0187 | 0.0294 | 0.0237 |
| K | 0.0288 | 0.0393 | 0.0234 | 0.0386 |
| P | 0.0018 | 0.0013 | 0.0009 | 0.0012 |
| O | 0.4716 | 0.4711 | 0.4823 | 0.4707 |
| Sum | 1 | 1 | 1 | 1 |

In order for the data to be in a statistically convenient form, Aitchison (1986) suggests to further transform the data in Table B2 by first forming a ratio with an arbitrary element (in the list) and then taking the logarithm. For the exemplary dataset this is shown in Table B3.

**Table B3: Log-ratio of element weight percentages, with respect to oxygen-wt%.**

| Sample | JT01 | JT02 | JT19 | JT20 |
|---|---|---|---|---|
| Elements | | | | |
| $\ln(Si/O)$ | -0.48531565 | -0.50379579 | -0.40607778 | -0.5042219 |
| $\ln(Ti/O)$ | -4.14567399 | -4.74687577 | -4.68001075 | -4.43477381 |
| $\ln(Al/O)$ | -1.72531 | -1.54064159 | -1.83464118 | -1.54183733 |
| $\ln(Fe/O)$ | -2.03494223 | -2.12384752 | -2.64974526 | -2.29276806 |
| $\ln(Mn/O)$ | -6.39894857 | -6.21033707 | -6.55719484 | -6.52451934 |
| $\ln(Mg/O)$ | -3.06839321 | -3.29293646 | -3.67672517 | -3.30896536 |
| $\ln(Ca/O)$ | -3.47357746 | -4.51287533 | -3.53142599 | -3.88207448 |
| $\ln(Na/O)$ | -3.31519303 | -3.22481996 | -2.79600952 | -2.98709658 |
| $\ln(K/O)$ | -2.79436382 | -2.48376691 | -3.0254978 | -2.50170175 |
| $\ln(P/O)$ | -5.56150947 | -5.90149576 | -6.28344485 | -5.99945492 |
| $\ln(O/O)$ | 0 | 0 | 0 | 0 |



The rationale behind this transformation is as follows. The division by an arbitrarily present element effectively transforms the space into an N-1-dimensional open space, where the parameters (i.e. ratios) may have values between 0 and ∞. The subsequent application of the logarithm further changes the space, such that the new parameters can have values between −∞ and ∞. This results in so-called log-ratios, which should ideally be following a multivariate normal distribution. As a consequence, we can calculate the mean log-ratio vector across all samples as well as its corresponding covariance matrix, which completely describes the multivariate normal distribution. In addition to these statistical parameters, the script "compo_analysis.py" outputs a graph that plots for all samples an order statistic, $z_r$, (see Aitchison, 1986). This enables us to visualise how different the data is from a multivariate normal distribution. If equal, they should fall on the red line, shown in Fig. B3.


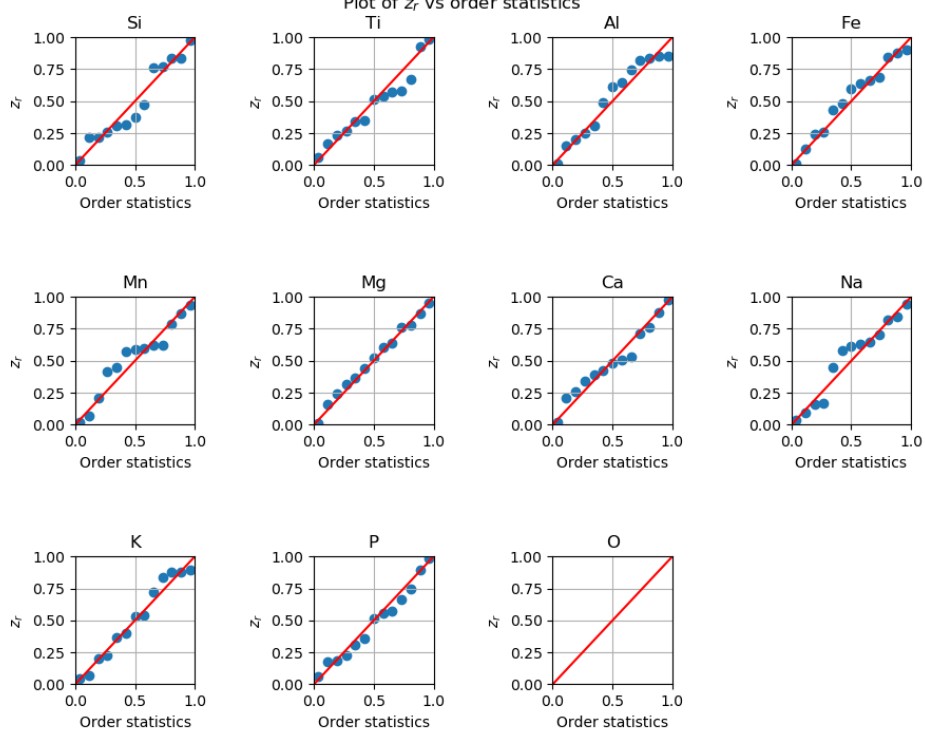


**Figure B3: Visual test for multivariate normality of the log-ratio data from Table B3 (This plot shows the full dataset, of which Table B3 is only an excerpt). Each subplot checks for marginal normality. Oxygen is the denominator variable (arbitrarily chosen) and does thus not appear in the plot.**






With a graph like Fig. B3 it is possible to check if the multivariate normal distribution is an appropriate model to describe the elemental composition data. For our example that we show in Fig. B3 this looks acceptable, with only slight deviations

for silicon, aluminium, manganese and sodium). Once the normality has been verified it is possible to generate random samples from this distribution. For every drawn sample it is then possible to calculate the weight percentages of the single elements by using the inverse formula to the log-ratio transformations

$$wt_{ele,i} = \frac{\exp(r_i)}{1 - \sum_{j=1}^{N_{ele}-1} \exp(r_j)}, \tag{B6}$$

for all numerator elements and

$$wt_{ele,N_{ele}} = \frac{1}{1 - \sum_{j=1}^{N_{ele}-1} \exp(r_j)} \tag{B7}$$

for the denominator element (here oxygen). In Eqs. (B6) and (B7) the $r_i$ denote the log-ratios from Table B3 and $N_{ele}$ is the total number of elements (in Table B2).

**B3 – Energy loss equation for rocks**

As stated in Eq. (7) the energy loss equation for rocks needs parameters that differ from the ones for pure elements. First, the

expression for density can directly be exchanged according to the density model (see Appendix B1). Second, it is possible to generate an expression for the average ionisation loss within a rock by exchanging three parameters. Density values that also enter within $\{a\}_{rock}$ can again be directly changed. The average $\{Z/A\}_{rock}$ may be exchanged with the elemental $Z/A$ by using

$$\{Z/A\}_{rock} = \sum_{i=1}^{N_{ele}} wt_{ele,i} * \frac{Z_i}{A_i}. \tag{B8}$$

$wt_{ele,i}$ are the weight fractions from Eqs. (B6) & (B7). Lastly, the mean excitation energy for the rock can be computed by

$$\ln\{I\}_{rock} = \frac{\sum_{i=1}^{N_{ele}} wt_{ele,i} * \frac{Z_i}{A_i} * \ln I_i}{\{Z/A\}_{rock}}. \tag{B9}$$

The radiation loss term, however, must be calculated as a weighted radiation energy loss over all elements. This means that the average can be written in a rather concise form,

$$\{b\}_{rock} = \sum_{i=1}^{N_{ele}} wt_{ele,i} * b_{ele,i}. \tag{B10}$$






**Appendix C – Metropolis Hastings technicalities**

This appendix chapter is a short summary of Gelman (2014, p. 284 – 287) and we refer to these pages for a detailed discussion of the calculations. This work presents a concept of how to assess the quality of a MCMC run. In particular, the aforementioned author proposes to analyse two quantities, the potential scale reduction factor $\hat{R}$ and the effective number of simulation draws $\hat{n}_{eff}$ for every parameter of interest. For every chain of a parameter the variance between different chains and within one chain is calculated. The posterior variance of the parameter is then estimated as a weighted average of these two types of variances. Finally, $\hat{R}$ is the quadratic ratio between the posterior variance and the variance within one chain. This quantity shows if the various chains have mixed or not, i.e. if they have explored the same region of the model space. If the posterior variance is much larger than the variances of the single chains, then the chains have not sufficiently explored the same region. Gelman (2014) propose to employ a threshold of 1.1 as a rule of thumb, below which the value of $\hat{R}$ would lie.

One problem that arises in MCMC algorithms is the inherent dependence of one simulation on the one before (this is the definition of a Markov chain). One considers that such a dependency does not introduce a bias if enough samples are drawn. However, this also means, that the effective, independent sample size is much smaller than the number of simulations. Therefore, Gelman (2014) proposes to calculate the effective number of simulation draws, $\hat{n}_{eff}$ in order to assess if one has enough independent samples The underlying idea here is to evaluate the correlations within the chains. An accepted threshold value for this parameter is $5m$, where $m$ is the number of sub-chains. For the calculation of $\hat{R}$ and $\hat{n}_{eff}$ the chains may be cut in half to generate more chains. Note, however, that $\hat{n}_{eff}$ can also be larger, which only means that the simulation standard error decreases. In our example we performed the calculations with two chains and a subdivision by 2, which means that our target quantity is around 20 ($= 5 * 4$). Most of our thickness parameters (i.e. cones with bedrock and ice; see Figs. 1 & 10) have, in fact, a $\hat{n}_{eff} > 100$, with only a few below.





**Appendix D – Construction of the smoothing kernel**

As stated in the main text, the user specifies the number of neighbouring pixels s to smooth over. The main idea is to
construct a roughly Gaussian smoothing kernel by approximating it with a binomial distribution. With help of the binomial
coefficient we can construct a vector of weights with $L = (2 * s + 1)$ entries. The weight vector is then given by

$$w_i = \frac{1}{2^{2*s}} \binom{L-1}{i},$$ (D1)

with $i \in \{0, .., L-1\}$. It is now possible to create a matrix by forming the dyadic product of $\vec{w}$ with itself, i.e.

$$K = \vec{w} \otimes \vec{w},$$ (D2)

or in index notation,

$$K_{ij} = w_i * w_j.$$ (D3)

As an example, we show how a smoothing kernel that smooths over two neighbouring pixels (i.e. $s = 2$) is constructed. This
is incidentally also the smoothing kernel we used to construct our ice-bedrock interface. The weight vector in this case is
given by

$$\vec{w} = \frac{1}{16} * (1 \quad 4 \quad 6 \quad 4 \quad 1).$$ (D4)

The weight vectors are, in fact, only the odd rows from Pascal's triangle, interpreted as vectors and normalised by a L1
norm. The smoothing matrix then takes the form

$$K = \frac{1}{256} \begin{pmatrix} 1 & 4 & 6 & 4 & 1 \\ 4 & 16 & 24 & 16 & 4 \\ 6 & 24 & 36 & 24 & 6 \\ 4 & 16 & 24 & 16 & 4 \\ 1 & 4 & 6 & 4 & 1 \end{pmatrix}.$$ (D5)






## Appendix E – Energy loss calculations for various elements and compounds

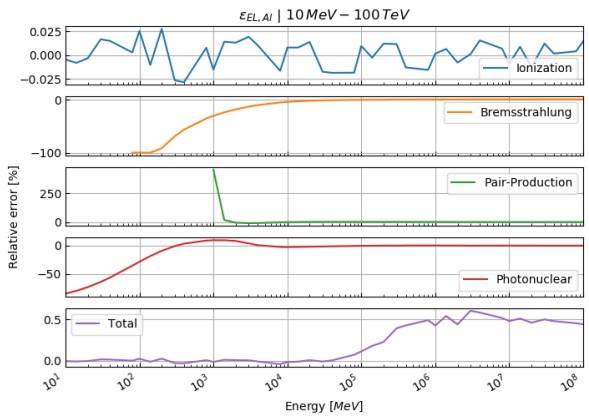
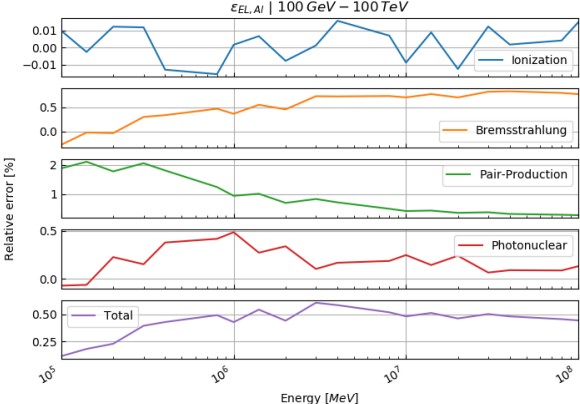

**Figure E1: Relative error of our energy loss calculations compared to the tabulated values from Groom et al. (2001) for aluminium in the energy ranges: (left) 10 MeV – 100 TeV, (right) 100 GeV – 100 TeV.**

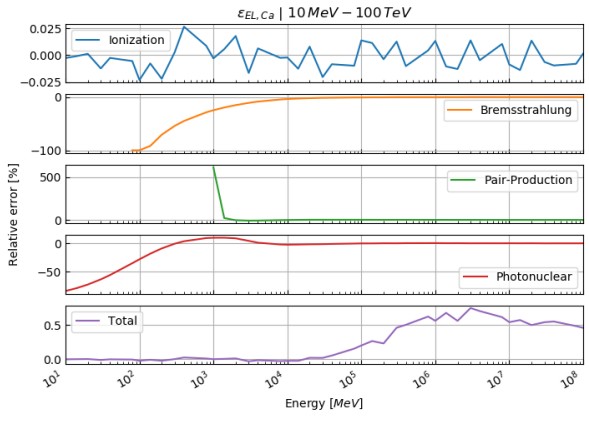
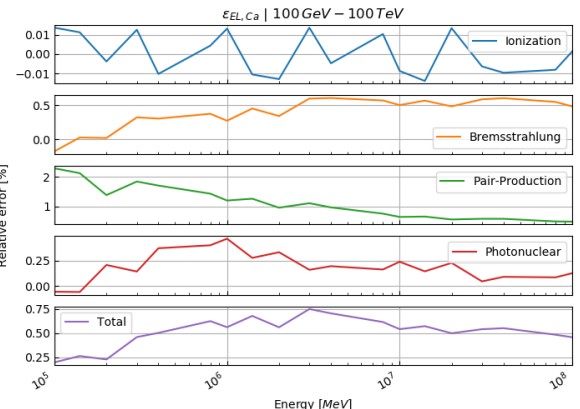

**Figure E2: Relative error of our energy loss calculations compared to the tabulated values from Groom et al. (2001) for calcium in the energy ranges: (left) 10 MeV – 100 TeV, (right) 100 GeV – 100 TeV.**





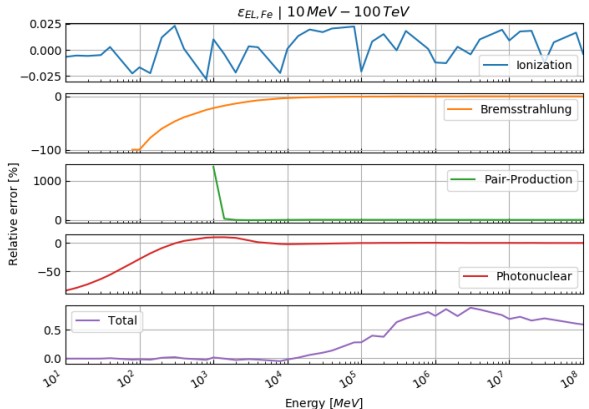
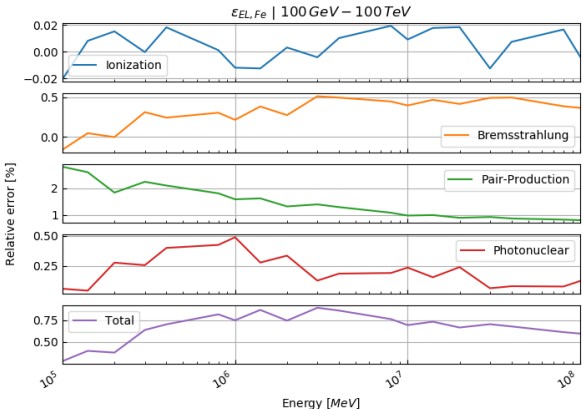

**Figure E3: Relative error of our energy loss calculations compared to the tabulated values from Groom et al. (2001) for iron in the energy ranges: (left) 10 MeV – 100 TeV, (right) 100 GeV – 100 TeV.**

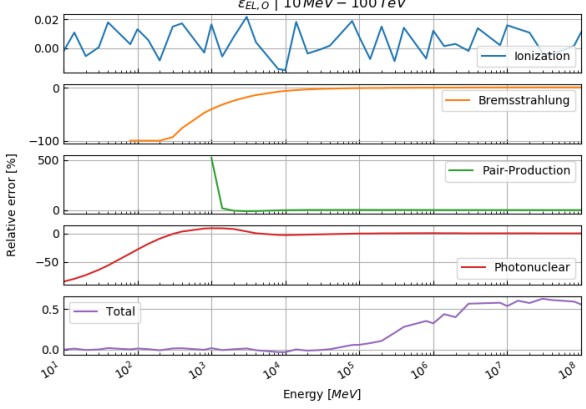
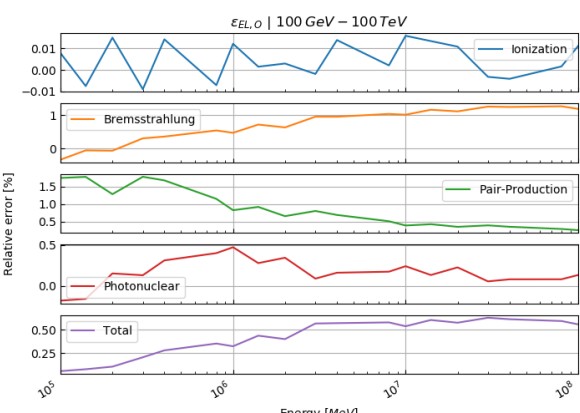

**Figure E4: Relative error of our energy loss calculations compared to the tabulated values from Groom et al. (2001) for oxygen in the energy ranges: (left) 10 MeV – 100 TeV, (right) 100 GeV – 100 TeV.**


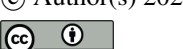



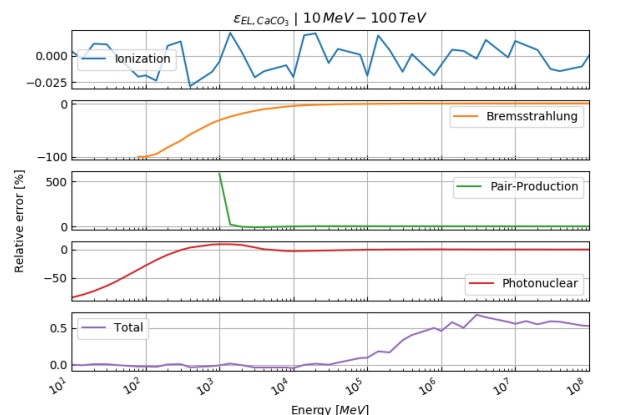

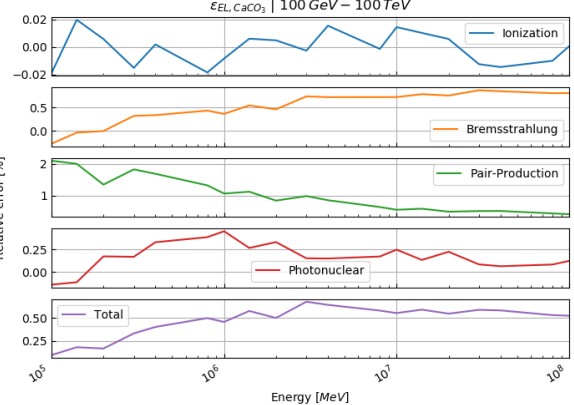

**Figure E6: Relative error of our energy loss calculations compared to the tabulated values from Groom et al. (2001) for calcium carbonate (calcite) in the energy ranges: (left) 10 MeV – 100 TeV, (right) 100 GeV – 100 TeV.**






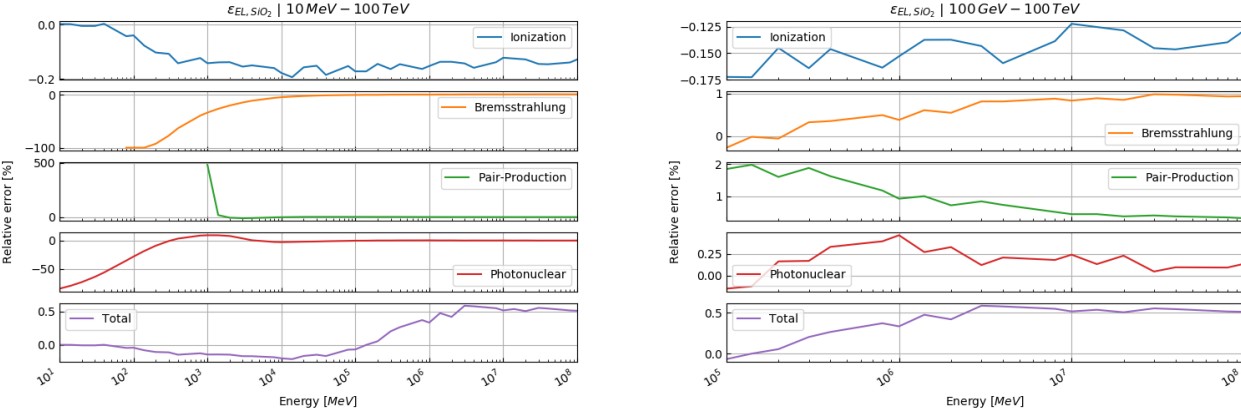

**Figure E7: Relative error of our energy loss calculations compared to the tabulated values from Groom et al. (2001) for silicon dioxide (quartz) in the energy ranges: (left) 10 MeV – 100 TeV, (right) 100 GeV – 100 TeV.**

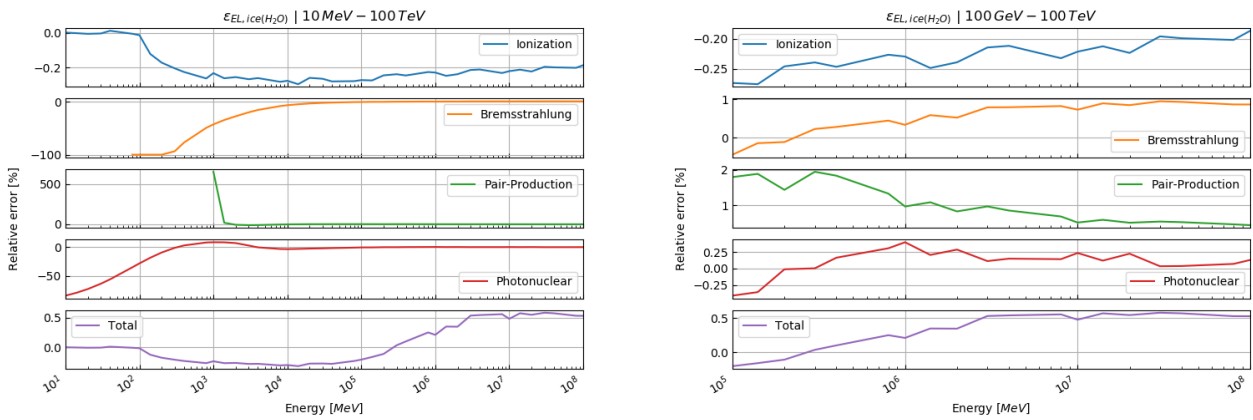

**Figure E8: Relative error of our energy loss calculations compared to the tabulated values from Groom et al. (2001) for ice in the energy ranges: (left) 10 MeV – 100 TeV, (right) 100 GeV – 100 TeV.**





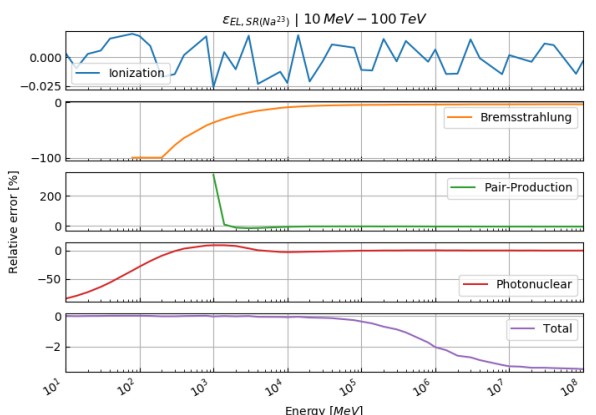
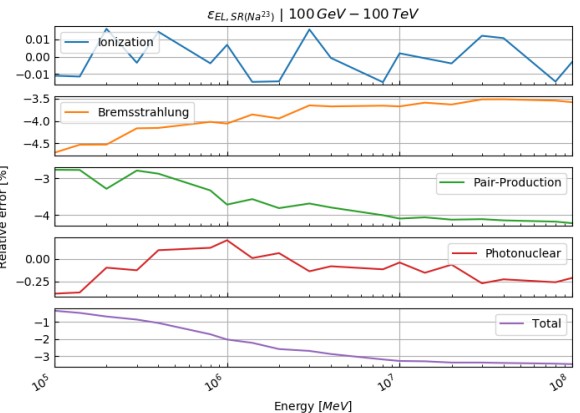

**Figure E9: Relative error of our energy loss calculations compared to the tabulated values from Groom et al. (2001) for standard rock ($^{23}_{11}$Na) in the energy ranges: (left) 10 MeV – 100 TeV, (right) 100 GeV – 100 TeV.**

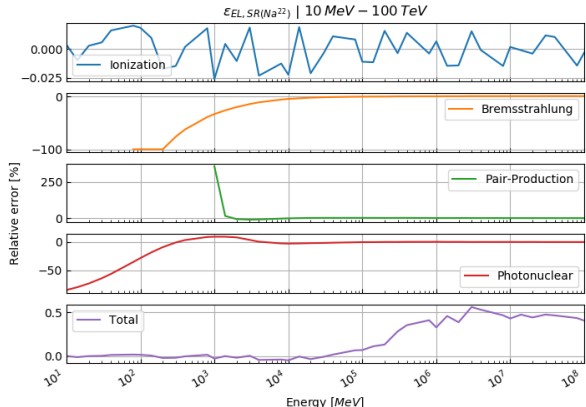
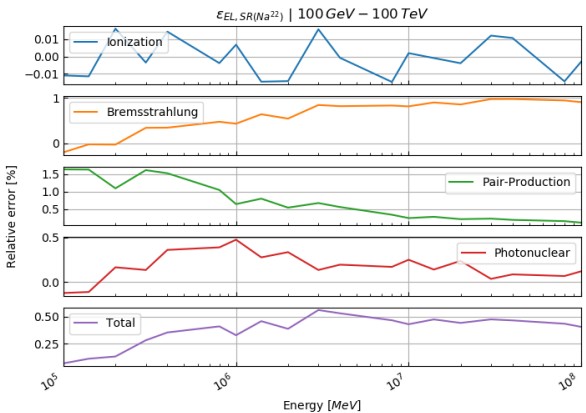

**Figure E10: Relative error of our energy loss calculations compared to the tabulated values from Groom et al. (2001) for standard rock ($^{22}_{11}$Na) in the energy ranges: (left) 10 MeV – 100 TeV, (right) 100 GeV – 100 TeV.**



**Code availability**

The source code of SMAUG 1.0 is publicly and freely available on *https://doi.org/10.5281/zenodo.5547356* (Lechmann et al., 2021b). The python packages required to run SMAUG are listed in the "requirements.txt" file.

**Data availability**

The data of the density and XRF measurements are included (i) in the files that can be downloaded from *https://doi.org/10.5281/zenodo.5547356* as well as (ii) in the supplementary material to this publication. The raw data from the Nishiyama et al. (2017) paper is publicly and freely available from the publisher's website.

**Author contributions**

AL, FS and AE designed the study

AL developed the code with contributions by MV, CP and RN

AL performed the numerical experiments with support by RN

DM and AL compiled geological data

AA, TA, PS, RN and CP verified the outcome of the numerical experiments

AL wrote the text with contributions from all co-authors

AL designed the figures with contributions by DM

All co-authors contributed to the discussion and finally approved the manuscript

**Competing interests**

The authors declare that they have no conflict of interest.

**Acknowledgements**

We thank the Swiss National Science Foundation (project No 159299 awarded to F. Schlunegger and A. Ereditato) for their financial support of this research project. Further, we want to thank the Jungfrau Railway Company for their continuing logistic support during our fieldwork in the central Swiss Alps. Finally, we want also to thank the High-Altitude Research Stations Jungfraujoch & Gornergrat for providing us with access to their research facilities and accommodation.





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
