# Peer review of "SMAUG v1.0 - a user-friendly muon simulator for the imaging of geological objects in 3D"

_Geoscientific Model Development, 2021_

## Referee Comment (RC2)

The article "SMAUG v1.0 – a user-friendly muon simulator for transmission tomography of geological objects in 3D" presents a code that allows performing inversion of muon flux measurement to reconstruct the density distribution of geological objects. The contribution of this paper with the code availability is of great interest as it could be used by geophysicists and geologists interested in the method. The paper clearly details the code functioning and its limits. The paper is well structured and well written. However some passages have to be clarified. Each symbol mentioned in the equations has also to be described. Thus I recommend a minor revision for this article.

The term "tomography" is usually used in geophysics to mention 2D images, often referring to vertical cross-sections. In the case of 3D reconstruction like here, the term "imagery" should be preferred in the title and in the text of the manuscript.

Before performing inversions and even acquiring data, geoscientists need a tool to evaluate the worth of muon imagery experiment. A virtual experiment could help them also to decide how to install muon sensors. They indeed need to know: How long do they have to install the sensors to detect a priori density contrasts ? Where should they install them to best capture the density distribution ? The tool you develop and present here might offer an opportunity to answer such questions. You could add a paragraph in the conclusion mentioning that.

The paper is rather long and I find the introduction quite vague when mentioning the inversion sought parameters. The muon imagery inversion aims at supplying the sounded medium density distribution. This should be clearly mentioned. You could place Fig. 3 earlier to help you explicitly indicate which are the input data required by the muon flux crossing the sensor forward model. Then discriminate which are the parameters sought by the inversion and which are the a priori parameters given either by field observation, geological knowledge, laboratory measurements or previous experimental analysis. Then remove part of the text in section 2 and 3 to avoid repetitions.

Replace the term ch. used in reference to chapter by section and sub-section.

The paper introduce a high number of variables some of them having the same symbol. Be aware to use only once each symbol. All symbols are not defined, pay attention to describe each of them in the text. All symbols could also be introduced in a table at the beginning of the paper.

l.16: "We address the need of the geoscientific community to participate in the data analysis"
I find this sentence quite condescending… Since the years 2010 geophysicists actively took part to the development of muon imagery and played with particles' models. By the way, you cite many of them… Geophysicists historically worked with physical models and performed inversions, one of them developed the basis of the inversion principles: Albert Tarantola… Rephrase this passage.

l. 55: repetition of "we aimed at"

Figure 1: I suggest you to rename the different plot a, b, c

L. 71: "physical parameters" be more precise and define the parameters sought by muon imagery

Figure 2: the quality of the figure has to be improved. Add to each module their inputs, outputs and how the modules interact with each other. Define also at the inversion step which are the sought parameters and which are the ones with a priori information. You could also enumerate the modules in the same order as in the text.

L. 90: "initial distribution of the lithologies" or a priori distribution…

L. 93: "to a set of parameters" → precise which ones

L. 104: "The interplay of these four submodules allows for the simulation of muon fluxes at the detector sites that are mostly located in an underground environment."
Use either the term module (I prefer this one) or submodule everywhere. Develop more precisely in the paragraph how the modules interplay.

L. 121: "As can be seen in Fig. 2, the inversion compares the simulated flux data with the measured ones. It also attempts to reduce the discrepancy between measurements and simulations by optimising the parameters in the simulation"
Inversion schemes aims at identifying the parameters of a model (or a set of modules) in order to reproduce the data observed given the a priori information available. Rewrite the sentence to be more precise.

L. 124: "As the mathematical optimization in muon tomography generally is nonlinear, one has to employ nonlinear solvers or even Monte Carlo techniques".
When working for producing a 3D block of density, the problems effectively turn non linear, a 2D tomography or else «radiography" doesn't require a MC inversion process.

L. 125: "one has to employ nonlinear solvers or even Monte Carlo techniques"
If finally you don't use such techniques, rewrite this sentence into something like → one classically employ nonlinear solvers or even Monte Carlo techniques

L. 128: "measurements from different sources"
Do you mean measurements acquired from different location ? Clarify

The introduction lacks a state of the art of the existing muon flux computation code. You only mention Geant4 and MUSIC. Without an extensive list of codes, cite the most used ones, describe their advantages and drawbacks to better highlight the supply of your code. Inversion tools to perform 3D muons imagery have also already been developed and applied, even in combination with other methods. However, the previous codes might not have been openly distributed as you do. Here are some suggested references:
- Bonnechi et al., 2015, A projective reconstruction method of underground or hidden structures using atmospheric muon absorption data, *Journal of Instrumentation*, *10*(02), P02003.
- Jourde et al., 2015, Improvement of density models of geological structures by fusion of gravity data and cosmic muon radiographies, Geosci. Instrum. Method. Data Syst. Discuss., 5, 83–116
- Niess et al., 2018, Backward Monte-Carlo applied to muon transport, Computer Physics Communications 229 (2018) 54–67
- Barnoud et al., 2019, Bayesian joint muographic and gravimetric inversion applied to volcanoes, Geophys. J. Int., 218, 2179–2194
- Lelièvre et al., 2019, Joint inversion methods with relative density offset correction for muon tomography and gravity data, with application to volcano imaging, *Geophysical Journal International*, *218*(3), 1685-1701.

L. 146: add a virgula after: "In muon tomography experiments"

Eq. 1: define: N, μ, i, sim and I

L.163: "$\Delta A_{eff,i}$ is solely dependent on the orientation of the bin"

but not on the geometry of the sensor ?

L. 169: "E cut,i describes the energy needed for a muon to enter the detector»
Doesn't it correspond to the energy required to cross the geological target and the muon sensor ?
This information is indeed given later, add it here for clarity.

L. 184: "and we reinterpret E cut,i as the minimum energy that is required to traverse the matter and to be registered at the detector".
To not loose the reader, give only one definition of that term in the text so give that one earlier when first mentioning it.

Eq. 5: define x

l. 273: "Even though these algorithms suffer from possible non-uniqueness solutions"
Inverse problems addressed with a Bayesian formulation might also suffer from non-uniqueness, but they should highlight this issue while descent algorithms or locally optimising algorithms might provide a local solution without warning on the existence of other solutions.

l. 284: "density values that were measured in the lab"
Be aware that measurements in the lab correspond to an elementary volume of rock that might not be representative of the macroscale sounded with muon imagery. The weathering of rock, the presence of fractures or faults might alter the medium density. This point has to be mentioned.

l.385: "As the total material thickness is known (detector position and digital elevation models are given), the sub-space containing the thickness parameter is endowed with the same mathematical structure as the one containing the composition parameter (i.e. one sum constraint), if the cone consists of more than just one segment."
Rewrite for clarity

l. 389: "within which we a-priori possess no information about the parameters"
Which parameters? Clarify

Equations 28 and 29: add parenthesis to explicit which terms are included in the sum or product on the $i$ or $j$ indices.

l. 409-410: "as for our problem it merely is a nuisance parameter" "which is of no particular interest but still has to be accounted for"
I would say that the muon flux computation is the key forward model, so I don't understand what you mean with such comments. Could you specify?

Equation 31: is the $i$ index lacking for $\rho$ and $c$ ?

l. 420: precise that eq. 32 is built also from eq. 12 and 25.

l. 421-424: This comment has to be considered with attention by scientists want to evaluate the interest of applying the method to their studied object. You could highlight it.

l. 434: "we retrieve $\tilde{\pi}(\rho rock)$, the posterior marginal pdf for the rock density"
Be aware that $\rho rock$ might be heterogeneous in the sounded medium due to weathering processes, presence of fractures and faults… You should already mention it here and discuss it later when you present your results.

l. 478: you used earlier the Σ symbol to represent a summation, use an other symbol here.

l. 478: explicit earlier the $J(0, c2\Sigma)$ term and precise which parameter is fixed to 0.

l. 481: which parameter does $r$ represent?

Fig. 4: add X and Y labels on the figure as well as the legend of the curves.

l. 535: the symbol "$r$" was used previously, maybe change the former.

l. 549: define H and k

l. 551: define Δx and Δy

l. 569: You used previously the symbol * for multiplication… maybe changed it in the previous equations

l. 588: "either from data" → from measurements?

l. 614: close the parenthesis

l. 614: "the energy loss calculations is based" → the energy loss calculations **are** based

l. 622: the error stands below 1%

Fig. 6 and 7: present here directly results for the materials you sound: ice (in dashed lines) and standard rock on the same figures.

Fusion Fig. 9 and 10 with dashed lines for the $^{22}_{11}Na$

Paragraph from l. 685: remove that paragraph that repeats what previously said.

Fusion Fig. 1 ans 11

l. 691: 2 grid pixel → clarify

l. 693: remove the sentence: "Figures 12 to 14 show the three cross-sections in detail"

paragraph from line 690: avoid writing m for metres in italic, the reader could think it represents a variable (notably as it is later used to represent the molar mass)

Fig. 12-14: Add the DEM +/- 2m error with dashed black lines to the figure

End of section 5.2: Discuss the variations between the DEM topography measurement and the topographic estimated from the measured muon flux damped to bedrock. As I mentioned earlier, the rocks sounded might be heterogeneous. As for rocks, the ice could also present some cracks, gullies, cavities… a void could also be present in between the rock upper limit and the bottom of the ice sheet. If you have prior information concerning that point, mention it. Add this discussion here.

l. 746: "In this study we have presented a model"
→ "In this study we have presented an inversion scheme"

In the conclusion, you could add a paragraph mentioning that the code you developed provide uncertainty estimates, which is very of particular interest.

l. 774: do not mix parameters and units in equations write better h0= a0+b0p (or other symbols). Then indicate that classically a0= 4900m and b0= 750mcGeV-1 and provide a reference. Then explain that you use the Nishiyama et al. (2017) suggested values.

Precise the appendix B title: "Rock parameters model"

B1 title → Density measurement

l. 781: "we constructed a density model by analysing various"
→ "we estimated the medium density by analysing various"

Fig. B1: strictly speaking a probability function integral should equal 1. Either normalize the represented values or rename the y label.

Fig. B2: add a legend with the different elements represented

B2 title → Composition of the medium

l. 863: add "the" before "following formula"

l. 879-881: divide the sentence for clarity

Fig. B3 you could remove the O plot and have 5x2 plots

l. 894: "For our example that we show"
→ "For the example shown"

l. 895: remove the lonely ending parenthesis

Equations B6 and B7: introduce all the variables in the text as well as the indices i and j

l. 910: & → and

Equation B9 and B10: introduce I and b

---

## Author Comment (AC1)

Line-by-line response to reviewer comment of Anonymous Referee #1

**Comment:**
*As a general comment, I would suggest to shorten the paper and move some technical part and reference to the python code in the Appendix.*
*For instance I would suggest to move Section 4 into an Appendix F.*
*I would also suggest to shorten Section 5 and focus on the geophysics only in this Section, since it is the main target of the paper. Section 5.1 may either be included into Section*
*2.2 in a shorter form, or move to the Appendix as well.*
**Response:**
We followed the suggestion of the reviewer and put sections 4 and 5.1 into the appendix, such that in the main text there should be a focus on mainly the Geophysics.

**Comment:**
*A Figure to explicit the different variables in Section 2.1 would be nice.*
**Response:**
We included a table showing all use variables in the main text

**Comment:**
*On the formal aspect, there are 2 conventions for the authors citations. Either : (Author et al, date) or : Author et al (date). Please unify.*
**Response:**
We checked all citations and made sure that they obey the style guideline of the journal where the "Author et al. (date)" format was used when the author is explicitly part of the sentence and the "(Author et al., date)"-style when the authors were not part of the sentence. We thereby also revised a few typos with the brackets.

**Comment:**
*l. 55 -> suppress "we aimed at".*
**Response:**
The second "we aimed at" has been eliminated

**Comment:**
*l. 220 -> "In order to..." -> incomplete sentence*
**Response:**
The sentence has been completed to:
"In order to speed up the computations – especially the calculation of the pair production cross-section, which includes two nested integrations – we utilise customised energy loss tables."

**Comment:**
*l. 404 -> "MCMC" not defined before*
**Response:**
Changed "MCMC" to Monte Carlo. In this way the term "MCMC" enters only when the Metropolis-Hastings algorithm is described.

**Comment:**
*l. 474 -> "Algorithm 1" -> there is no 2*
**Response:**
We omitted the 1.

---

## Author Comment (AC2)

Line-by-line response to reviewer comment of Dr. Nolwenn Lesparre

**Comment:**

*The term "tomography" is usually used in geophysics to mention 2D images, often referring to vertical cross-sections. In the case of 3D reconstruction like here, the term "imagery" should be preferred in the title and in the text of the manuscript.*

**Response:**

As stated in the general response to the reviewer comment we adopt the notion of "imaging" in the title and on occasions where we refer to the 3D reconstruction. In the main text, muon tomography is introduced as a geophysical imaging technique. Additionally, "tomography" is used as part of the term "muon tomography" which is a (admittedly unprecise) description of the technology and widely used in the community. We therefore find it better to leave this term as it is.

**Comment:**

*Before performing inversions and even acquiring data, geoscientists need a tool to evaluate the worth of muon imagery experiment. A virtual experiment could help them also to decide how to install muon sensors. They indeed need to know: How long do they have to install the sensors to detect a priori density contrasts ? Where should they install them to best capture the density distribution ? The tool you develop and present here might offer an opportunity to answer such questions. You could add a paragraph in the conclusion mentioning that.*

**Response:**

We added the following paragraph:

"In its current state, SMAUG may be of help to researchers a) who plan to use muon tomography in their own research, such that the feasibility of the use of this technology can be evaluated in a virtual experiment, b) who want to use a submodule for the analysis of their own muon tomography, or c) plan to perform a subsurface interface reconstruction similar to our study. ""

**Comment:**

*The paper is rather long and I find the introduction quite vague when mentioning the inversion sought parameters. The muon imagery inversion aims at supplying the sounded medium density distribution. This should be clearly mentioned. You could place Fig. 3 earlier to help you explicitly indicate which are the input data required by the muon flux crossing the sensor forward model. Then discriminate which are the parameters sought by the inversion and which are the a priori parameters given either by field observation, geological knowledge, laboratory measurements or previous experimental analysis. Then remove part of the text in section 2 and 3 to avoid repetitions.*

**Response:**

Concerning the sought parameters in the inversion, we have now added a subsection in Sect. 2, describing the choice of parametrisation. Fig. 2 now also highlights which parameters are important (i.e. density and thicknesses of the segments), next to where information on the various parameters is added. Moreover, we indicated in the list of parameters the ones that are of ultimate interest to us.

**Comment:**

*Replace the term ch. used in reference to chapter by section and sub-section.*

**Response:**

Changed "chapter" to "section" throughout the text.

**Comment:**

*The paper introduce a high number of variables some of them having the same symbol. Be aware to use only once each symbol. All symbols are not defined, pay attention to describe each of them in the text. All symbols could also be introduced in a table at the beginning of the paper.*

**Response:**

We have reviewed all the parameters in the main text and a) changed their symbols (to avoid same symbols). Further we provide now a table (Table 1), which lists the new parameters. Because the list is rather long, we grouped the parameters into sections where they appear first.

**Comment:**
l.16: "We address the need of the geoscientific community to participate in the data analysis"
I find this sentence quite condescending… Since the years 2010 geophysicists actively took part to the development of muon imagery and played with particles' models. By the way, you cite many of them… Geophysicists historically worked with physical models and performed inversions, one of them developed the basis of the inversion principles: Albert Tarantola… Rephrase this passage.

**Response:**
It was never our intention to be condescending and we acknowledge that many geoscientists played very important roles in driving this technology forward. Nevertheless, we also see that many geoscientists who would use muon tomography as a mere tool (not as their primary research) have a certain amount of reservation when it comes to utilising this technology in their own research. As muon tomography is a "relatively" new geophysical technology, not much experience is present. Therefore it might seem risky to use this technology in one's own research (in the end also researchers need to perform a cost-risk analysis of their research). Thus, we see our task in showing how this technology may be used, and we thus provide tools to gather experience. It is our conviction that by making the knowledge better accessible for our fellow geoscientists we may convince some of them to employ muon tomography in their research.

Nevertheless, we see that our sentence may be offensive. We have rewritten that part to:

"As the data analysis is still mostly done by particle physicists, much of the know-how is concentrated in particle physics and specialised geophysics institutes. SMAUG, a toolbox consisting of several modules that cover the various aspects of data analysis in a muon tomographic experiment, aims at providing access to a structured data analysis framework. The goal is to make muon tomography more accessible to a broader geoscientific audience."

**Comment:**
l. 55: repetition of "we aimed at"
**Response:**
Eliminated surplus "we aimed at"

**Comment:**
Figure 1: I suggest you to rename the different plot a, b, c
**Response:**
Done

**Comment:**
L. 71: "physical parameters" be more precise and define the parameters sought by muon imagery
**Response:**
Added "(usually density and/or the thickness of a part of the material)" to specify the general situation

**Comment:**
Figure 2: the quality of the figure has to be improved. Add to each module their inputs, outputs and how the modules interact with each other. Define also at the inversion step which are the sought parameters and which are the ones with a priori information. You could also enumerate the modules in the same order as in the text.
**Response:**
Figure 2 now shows a detailed version of the general sketch from before. It is also shown schematically what the basic principle of the inversion is and how exactly simulation and experiment are tied into it. We additionally indicated where the laboratory measurement and digital elevation model information is integrated.

**Comment:**
*L. 90: "initial distribution of the lithologies" or a priori distribution…*
**Response:**
At this stage we do not specify a probabilistic model of the lithologies. The word "distribution" is not to be interpreted in a statistical sense but a spatial one. We have to specify an initial model which describes how the materials are distributed. As such it is only one instance of a statistical distribution. In order to clarify this unlucky double meaning of the word distribution, we specified that we mean "spatial distribution"

**Comment:**
*L. 93: "to a set of parameters" → precise which ones*
**Response:**
Added (namely thickness, density, and composition) to specify

**Comment:**
*L. 104: "The interplay of these four submodules allows for the simulation of muon fluxes at the detector sites that are mostly located in an underground environment."*
*Use either the term module (I prefer this one) or submodule everywhere. Develop more precisely in the paragraph how the modules interplay.*
**Response:**
The whole second part of Sect. 1.1 has been rewritten to form a narrative similar to the flow in Fig. 2. We also took this as an opportunity to better highlight the connections between the modules.

**Comment:**
*L. 121: "As can be seen in Fig. 2, the inversion compares the simulated flux data with the measured ones. It also attempts to reduce the discrepancy between measurements and simulations by optimising the parameters in the simulation"*
*Inversion schemes aims at identifying the parameters of a model (or a set of modules) in order to reproduce the data observed given the a priori information available. Rewrite the sentence to be more precise.*
**Response:**
Made sentence clearer:
"As can be seen in Fig. 2, the inversion compares the simulated flux data with the measured ones. This problem is solved by finding the set of parameters (material density and the thicknesses of the overlying materials), that adhere to the constraints of the available a priori information and minimise the aforementioned discrepancy between measurement and simulation."

**Comment:**
*L. 124: "As the mathematical optimization in muon tomography generally is nonlinear, one has to employ nonlinear solvers or even Monte Carlo techniques".*
*When working for producing a 3D block of density, the problems effectively turn non linear, a 2D tomography or else «radiography" doesn't require a MC inversion process.*
**Response:**
In this sentence we refer to the nonlinearities in the energy loss equation. In general the energy loss equation is (mildly) non-linear in the density as the ionisation losses contain terms that include density information which can not be separated. This was the original motivation to use Monte Carlo inversion. MC techniques are, however, also enough versatile to tackle nonlinearities due to the parametrisation. We have adapted our sentence to:
"As the energy loss equation in general is nonlinear, also the mathematical optimisation in muon tomography is nonlinear. This is classically solved by either a linearisation of the physical equations or by employing nonlinear solvers. A further difficulty is introduced when working in 3D. Monte Carlo techniques are, however, enough versatile to address these challenges, which is our main motivation for working with them."

**Comment:**

*L. 125: "one has to employ nonlinear solvers or even Monte Carlo techniques"*
*If finally you don't use such techniques, rewrite this sentence into something like → one classically employ nonlinear solvers or even Monte Carlo techniques*

**Response:**

see response above

**Comment:**

*L. 128: "measurements from different sources"*
*Do you mean measurements acquired from different location ? Clarify*

**Response:**

Added explaining parentheses
(i.e. muon flux measurement, laboratory, geological field measurements, maps, etc.)

**Comment:**

*The introduction lacks a state of the art of the existing muon flux computation code. You only mention Geant4 and MUSIC. Without an extensive list of codes, cite the most used ones, describe their advantages and drawbacks to better highlight the supply of your code. Inversion tools to perform 3D muons imagery have also already been developed and applied, even in combination with other methods. However, the previous codes might not have been openly distributed as you do. Here are some suggested references:*

- *Bonnechi et al., 2015, A projective reconstruction method of underground or hidden structures using atmospheric muon absorption data, Journal of Instrumentation, 10(02), P02003.*
- *Jourde et al., 2015, Improvement of density models of geological structures by fusion of gravity data and cosmic muon radiographies, Geosci. Instrum. Method. Data Syst. Discuss., 5, 83–116*
- *Niess et al., 2018, Backward Monte-Carlo applied to muon transport, Computer Physics Communications 229 (2018) 54–67*
- *Barnoud et al., 2019, Bayesian joint muographic and gravimetric inversion applied to volcanoes, Geophys. J. Int., 218, 2179–2194*
- *Lelièvre et al., 2019, Joint inversion methods with relative density offset correction for muon tomography and gravity data, with application to volcano imaging, Geophysical Journal International, 218(3), 1685-1701.*

**Response:**

We followed the suggestion of the reviewer and provided a short overview of the suggested works and how these motivate us to provide our own code framework. We thus modified Sect. 1.2.

**Comment:**

*L. 146: add a virgula after: "In muon tomography experiments" Eq. 1: define: N, μ, i, sim and I*

**Response:**

Comma added, $N, \mu, i$ defined. "I" has been replaced by $\Phi$, which is defined in Eq. (5). $sim$ is no longer needed.

**Comment:**

*L.163: "ΔA eff,i is solely dependent on the orientation of the bin" but not on the geometry of the sensor ?*

**Response:**

Normally, the detectors that are used are flat (i.e. not curved), in which case the statement is true. We added "(as long as the detector records muons on flat surfaces)" to strengthen our argument

**Comment:**

*L. 169: "E cut,i describes the energy needed for a muon to enter the detector»*
*Doesn't it correspond to the energy required to cross the geological target and the muon sensor? This information is indeed given later, add it here for clarity.*

**Response:**

This is correct. We added a respective passage.

**Comment:**

*L. 184: "and we reinterpret E cut,i as the minimum energy that is required to traverse the matter and to be registered at the detector".*
*To not loose the reader, give only one definition of that term in the text so give that one earlier when first mentioning it.*

**Response:**

We revised this part and eliminated the second mention of the definition of $E_{cut,i}$. We also omitted the introduction of $E_0$ until Eq. (12) where it is first used and defined.

**Comment:**

*Eq. 5: define x*

**Response:**

Added definition

**Comment:**

*l. 273: "Even though these algorithms suffer from possible non-uniqueness solutions"*
*Inverse problems addressed with a Bayesian formulation might also suffer from non-uniqueness, but they should highlight this issue while descent algorithms or locally optimising algorithms might provide a local solution without warning on the existence of other solutions.*

**Response:**

Changed the sentence to:

« One difficulty of such algorithms is that in the case of non-unique solutions (which occur when there are local minima that might be a solution to the optimisation) the user has no constraints to infer if a local or the desired global minimum has been reached. A further problem of descent methods is the calculation of the derivatives of the forward model with respect to the parameter values."

We added also a sentence to convey the message of non-unique Bayesian solutions:

"We have to add that Bayesian methods do not solve the non-uniqueness problem, but they provide the user with enough information to spot these local solutions of the optimisation."

**Comment:**

*l. 284: "density values that were measured in the lab"*
*Be aware that measurements in the lab correspond to an elementary volume of rock that might not be representative of the macroscale sounded with muon imagery. The weathering of rock, the presence of fractures or faults might alter the medium density. This point has to be mentioned.*

**Response:**

We agree that this problem always exists when analysing surface samples and inferring the same density for the whole rock body. In our experience, however, when enough samples are considered from different locations, one could potentially better see these outliers. We added a paragraph in Appendix B1 to hint at this issue:

"One word of warning has to be made here. The measured densities of rock might be affected with a systematic error. Namely, the rock samples that are analysed were all gathered from surface near locations (in our case inside the tunnel or outside, i.e. where rocks are accessible). This means that they could have been subject to weathering processes that alter the density of the rock in such a way that the samples are not representative of the whole rock body anymore. Possible countermeasures would be to compare drilled samples from deeper within the rock body with the surface samples, etc."

**Comment:**

*l.385: "As the total material thickness is known (detector position and digital elevation models are given), the sub-space containing the thickness parameter is endowed with the same mathematical structure as the one containing the composition parameter (i.e. one sum constraint), if the cone consists of more than just one segment."*
*Rewrite for clarity*

**Response:**

Rewrote:

"We know the total thickness, due to our information on the detector position and the surface position from digital elevation models. Thus we have, equal to the compositions' weight fractions (that add up to 1), a sum constraint (i.e. the sum of lengths of all segments must equal the total distance from the detector to the surface). The mathematical structure of the parameter sub-space is consequently also the same."

**Comment:**

*l. 389: "within which we a-priori possess no information about the parameters" Which parameters? Clarify*

**Response:**

Here we talk about the thickness parameters. We slightly rewrote the sentence to make it clearer:

"One can therefore safely assume that the thickness parameters can be presented in a log-ratio space, within which we a-priori possess no additional information. Thus, we attribute the thickness parameters a multidimensional uniform distribution within the log-ratio space."

**Comment:**

*Equations 28 and 29: add parenthesis to explicit which terms are included in the sum or product on the i or j indices.*

**Response:**

Done

**Comment:**

*l. 409-410: "as for our problem it merely is a nuisance parameter" "which is of no particular interest but still has to be accounted for"*
*I would say that the muon flux computation is the key forward model, so I don't understand what you mean with such comments. Could you specify?*

**Response:**

With that sentence we mean that even though the calculation of the muon flux is still key, we do not want to treat it as an explicit parameter in the inversion that has to be simulated by the MCMC later on. We are not interested in the resolution of the muon flux. Instead we want to know how the energy loss calculation relates directly to the number of muons. We included this statement in the text:

"We first get rid of the flux parameter, as for our problem it merely is a nuisance parameter. This is an official term for a parameter in the inversion, which is of no particular interest but still has to be accounted for. Specifically, we mean that even though the calculation of the muon flux is important, we do not want to treat it as an explicit parameter that is simulated by the code. To achieve this, we integrate over all possible values of the muon flux, $\vec{f}$ within its uncertainty and we can relate the results of the energy loss calculation (encoded in $\mu_{f_i}$; see Eq. 29) directly to the measured number of muons, $d_i$."

**Comment:**
*Equation 31: is the i index lacking for ρ and c ?*
**Response:**
We used a model where $\rho$ and $c$ are the same parameters for every cone. In the DAG (Fig. 3) those parameters are outside the "cones". They consequently do not depend on the index $i$. Later we use a special model (the "SICOBI", i.e. single-cone-bin inversion), which introduces this dependence explicitly. However, not in Eqs. (32) & (33).

**Comment:**
*l. 420: precise that eq. 32 is built also from eq. 12 and 25.*
**Response:**
Done

**Comment:**
*l. 421-424: This comment has to be considered with attention by scientists want to evaluate the interest of applying the method to their studied object. You could highlight it.*
**Response:**
We referred to an earlier work of ours where we give an example how this can be quickly estimated.

**Comment:**
*l. 434: "we retrieve $\tilde{\pi}(\rho_{rock})$, the posterior marginal pdf for the rock density"*
*Be aware that might be heterogeneous in the sounded medium due to weathering processes, presence of fractures and faults… You should already mention it here and discuss it later when you present your results.*
**Response:**
This might be true. However, as we used the homogeneity of the material parameters already in the earlier study (Nishiyama et al., 2017). Since we want to verify our new code, we cannot change that assumption as it might yield different results. We added a paragraph in Sect. 2.4 raising awareness for this circumstance and explaining our choice.

**Comment:**
*l. 478: you used earlier the Σ symbol to represent a summation, use an other symbol here.*
**Response:**
Changed $\Sigma$ to $S$.

**Comment:**
*l. 478: explicit earlier the (0, 2Σ) term and precise which parameter is fixed to 0.*
**Response:**
Added explanation of the function arguments in the algorithm description.

**Comment:**
*l. 481: which parameter does r represent?*
**Response:**
Renamed $r$ to $O$ and named it "odds ratio". In Bayesian calculations the relative probability between two models is given as an odds ratio.

**Comment:**
*Fig. 4: add X and Y labels on the figure as well as the legend of the curves.*
**Response:**
Done

**Comment:**
*l. 535: the symbol "r" was used previously, maybe change the former.*
**Response:**
$r$ from before has been changed to $O$. As $r$ is still used later on as an index we changed the row and column numbers to $N_{row}$ and $N_{col}$.

**Comment:**
*l. 549: define H and k*
**Response:**
$H_i$ has been defined now and $k$ has been substituted with $i$ as it is a sum over all cones

**Comment:**
*l. 551: define Δx and Δy*
**Response:**
$\Delta x_i$ and $\Delta y_i$ are now also defined.

**Comment:**
*l. 569: You used previously the symbol \* for multiplication… maybe changed it in the previous equations*
**Response:**
Changed $*$ to $\circledast$ to describe the convolution

**Comment:**
*l. 588: "either from data" → from measurements?*
**Response:**
Changed it to «from measurements"

**Comment:**
*l. 614: close the parenthesis*
*l. 614: "the energy loss calculations is based" → the energy loss calculations are based*
**Response:**
No longer needed as the verification of the energy loss model has been moved to Appendix E and the introduction of the verification section has been shortened.

**Comment:**
*l. 622: the error stands below 1%*
**Response:**
Corrected to "the relative error is generally below 1%"

**Comment:**
*Fig. 6 and 7: present here directly results for the materials you sound: ice (in dashed lines) and standard rock on the same figures.*
**Response:**
The figures for the energy loss verification have been moved to Appendix E. There we already provide the necessary figures for ice. Moreover, as ice is a tabulated compound in Groom et al. (2001), the behaviour of the energy loss calculations is expected to be similar to those of other listed compounds. Within 0.25% this is true. The main problem here really was with the definition of standard rock that had a discrepancy of a few %. We thus do not see a need to change the figures here.

**Comment:**

*Fusion Fig. 9 and 10 with dashed lines for the $^{22}_{11}Na$*

**Response:**

We have reorganised Appendix E1 & E2 such that the standard rock calculations are next to each other. We find it still helpful for these figures to remain in their original form such that they may be better comparable to the other figures in Appendix E.

We have however omitted the figure doubles of the standard rock calculation. Every image is now unique.

**Comment:**

*Paragraph from l. 685: remove that paragraph that repeats what previously said.*

**Response:**

Not needed anymore as the introduction to Sect. 4 (Model Verification) has been omitted and the section on the energy loss calculation has been moved to Appendix E. As such the introduction to the verification of the bedrock-ice interface reconstruction is now not repetitive anymore.

**Comment:**

*Fusion Fig. 1 ans 11*

**Response:**

We merged the overview figure with Fig. 1, which acts now twofold as overview map and as an explanation of the parts of the reconstruction (i.e. cross-sections, etc.)

**Comment:**

*l. 691: 2 grid pixel → clarify*

**Response:**

Added an explanation that ties back to the smoothing matrix formalism in Eq. (43)

**Comment:**

*l. 693: remove the sentence: "Figures 12 to 14 show the three cross-sections in detail"*

**Response:**

Done

**Comment:**

*paragraph from line 690: avoid writing m for metres in italic, the reader could think it represents a variable (notably as it is later used to represent the molar mass)*

**Response:**

We revised the whole manuscript and changed every reference to a "variable" to italic whereas every reference to a "physical unit" is now in upright roman. This should now be in accordance with the journals guidelines.

**Comment:**

*Fig. 12-14: Add the DEM +/- 2m error with dashed black lines to the figure*

**Response:**

The systematic error has been directly propagated to the reconstructed surface. We think that when the error is explicitly shown on the DEM there are too many intersecting lines on the plot that might obscure the focus of our result. Thus, we prefer to leave the figures as they are now.

**Comment:**

*End of section 5.2: Discuss the variations between the DEM topography measurement and the topographic estimated from the measured muon flux damped to bedrock. As I mentioned earlier, the rocks sounded might be heterogeneous. As for rocks, the ice could also present some cracks, gullies, cavities… a void could also be present in between the rock upper limit and the bottom of the ice sheet. If you have prior information concerning that point, mention it. Add this discussion here.*

**Response:**

We quickly discussed the over-/underestimation of the DEM in the bedrock region as either a smoothing effect or due to heterogeneities (fractures, weathering). As we do not possess conclusive data we left the decision open for future studies. However, as the agreement of the reconstructions is solid we leave it as it is. Note that in this study the goal was to compare our reconstruction algorithm with the calculations done by Nishiyama et al. (2017), thus we have to employ the same additional inversion procedures (i.e. damping, smoothing).

**Comment:**

*l. 746: "In this study we have presented a model"*
*→ "In this study we have presented an inversion scheme"*

**Response:**

Changed

**Comment:**

*In the conclusion, you could add a paragraph mentioning that the code you developed provide uncertainty estimates, which is very of particular interest.*

**Response:**

Changed : "The propagation of uncertainties thus occurs automatically within this formalism." to "The propagation of uncertainties thus occurs automatically within this formalism, providing uncertainty estimates on all parameters of interest."

**Comment:**

*l. 774: do not mix parameters and units in equations write better h0= a0+b0p (or other symbols). Then indicate that classically a0= 4900m and b0= 750mcGeV-1 and provide a reference. Then explain that you use the Nishiyama et al. (2017) suggested values.*

**Response:**

We have now omitted the units in these formulae

**Comment:**

*Precise the appendix B title: "Rock parameters model"*

**Response:**

With the newly arranged Fig. 2, this should not be necessary anymore. There it is stated that the rock model consists of a density and a composition model

**Comment:**

*B1 title → Density measurement*

**Response:**

This is an example output from SMAUG. As such the title is custom made for each subsample. As a matter of fact, this plot does not show the measurement of the density explicitly, but a Monte-Carlo uncertainty propagation of all involved measured parameters to the material density. Therefore, the title Histogram should be fine.

**Comment:**
*l. 781: "we constructed a density model by analysing various"*
*→ "we estimated the medium density by analysing various"*
**Response:**
Changed to « we estimated the density of the lithology by analysing various…"

**Comment:**
*Fig. B1: strictly speaking a probability function integral should equal 1. Either normalize the represented values or rename the y label.*
**Response:**
This is the case. The area under a Gaussian is roughly
$A = \frac{H*FWHM}{2.35*0.3989}$, where H is the height of the Gaussian and FWHM its full-width-half-maximum. By plugging in some rough numbers (H ~ 24 and FWHM ~ 0.04 (= 2.705-2.665)) one gets an area of A ~1.02. This should be fine.

**Comment:**
*Fig. B2: add a legend with the different elements represented B2 title → Composition of the medium*
**Response:**
We added the legend of this figure and added also the units to the axes. The title however is left as before as a) this is still a summary of a material density and b) the idea is to perform a kernel density estimate of the material density.

**Comment:**
*l. 863: add "the" before "following formula"*
**Response:**
Done

**Comment:**
*l. 879-881: divide the sentence for clarity*
**Response:**
We dropped the last part of the sentence making it clearer:
"As a consequence, we can describe this distribution by the mean log-ratio vector across all samples as well as its corresponding covariance matrix."

**Comment:**
*Fig. B3 you could remove the O plot and have 5x2 plots*
**Response:**
We perefer to keep the O in the plot, as a safeguard against a miscalculation. Because in the log-ratio space one element will be the "denominator variable" (as oxygen in Table B6). As this can be arbitrarily chosen it still emerges in this example output of the code.

**Comment:**
*l. 894: "For our example that we show"*
*→ "For the example shown"*
**Response:**
Changed

**Comment:**
*l. 895: remove the lonely ending parenthesis*
**Response:**
Done

**Comment:**

*Equations B6 and B7: introduce all the variables in the text as well as the indices i and j l. 910: & → and*

**Response:**

Changed the text after the equation to:

"for the denominator element (here oxygen). In Eqs. (B6) and (B7) the $r_i$ denote the log-ratios from Table B3, $N_{ele}$ is the total number of elements (in Table B2) and the index $i$ runs through all elements (e.g. in Table B5) and the index $j$ runs through all elements except the denominator variable. "

**Comment:**

*Equation B9 and B10: introduce I and b*

**Response:**

We added the relevant variables in the text right next to the explanations:

"Lastly, the mean excitation energy, $\{I\}_{rock}$, for the rock can be computed by…"

"The radiation loss term, $\{b\}_{rock}$, must be calculated…"

---

## Author Response (AR2)

Dear Editor,

We thank you for your helpful inputs to further improve our Figure 2. We heeded your advice and changed the colour coding of the module titles, such that also people who have problems to distinguish between red and green can read the figure. Moreover, we added also the titles in the "Inversion module" to further improve its readability. Finally, as we reduced the white space of the figure, we felt that we should also hint at the outputs of the modules to retain the most important information in one view.

Further, we changed the symbol for the Odds ratio from $O$ to $R$. It should now not conflict with other symbols anymore.

In addition to the revised Figure 2, we also changed the coloured cells in Table 1. We were pointed out by the technical check staff that this is impossible. So we decided to highlight the target parameters in a bold font.

There table numbering in Appendix B started from B2, which we corrected.

Kind regards,

Alessandro Lechmann